# ssDNA accessibility of Rad51 is regulated by orchestrating multiple RPA dynamics

Jiawei Ding[1,6], Xiangting Li [2,6], Jiangchuan Shen [3,6], Yiling Zhao[1,5], Shuchen Zhong[4], Luhua Lai [1,4], Hengyao Niu [3] ✉ & Zhi Qi [1] ✉

The eukaryotic single-stranded DNA (ssDNA)-binding protein Replication Protein A (RPA) plays a crucial role in various DNA metabolic pathways, including DNA replication and repair, by dynamically associating with ssDNA. While the binding of a single RPA molecule to ssDNA has been thoroughly studied, the accessibility of ssDNA is largely governed by the bimolecular behavior of RPA, the biophysical nature of which remains unclear. In this study, we develop a three-step low-complexity ssDNA Curtains method, which, when combined with biochemical assays and a Markov chain model in non-equilibrium physics, allow us to decipher the dynamics of multiple RPA binding to long ssDNA. Interestingly, our results suggest that Rad52, the mediator protein, can modulate the ssDNA accessibility of Rad51, which is nucleated on RPA coated ssDNA through dynamic ssDNA exposure between neighboring RPA molecules. We find that this process is controlled by the shifting between the protection mode and action mode of RPA ssDNA binding, where tighter RPA spacing and lower ssDNA accessibility are favored under RPA protection mode, which can be facilitated by the Rfa2 WH domain and inhibited by Rad52 RPA interaction.

Replication Protein A (RPA) is the primary single-stranded DNA (ssDNA)-binding protein in eukaryotes, and it plays a crucial role in coordinating downstream DNA metabolic pathways, such as DNA replication and repair, because it functions as the first responder to exposure of ssDNA, which is the key intermediates for these processes[1–5]. Mutations of RPA are associated with various human diseases, including breast and colon cancer[6]. During DNA replication, RPA is involved in initiation, lagging strand synthesis, and replication checkpoint activation. In DNA repair, RPA is necessary for almost all repair pathways, including mismatch repair, nucleotide excision repair, and homologous recombination (HR) mediated DNA double-strand break repair[2,5]. While both RPA-interacting proteins (RIPs) and post-translational modifications (PTMs) can regulate the function of RPA[3], the precise mechanism by which RPA balances DNA protection

with the activation of downstream DNA metabolic pathways remains unclear. Notably, the same ssDNA exposed during lagging strand synthesis is also subjected to HR, which may generate DNA catenanes and cause genome instability.

Like its human counterpart, RPA (Replication Protein A, or Replication Factor A) from the budding yeast, *Saccharomyces cerevisiae*, is a heterotrimer of three subunits, viz. Rfa1, Rfa2, and Rfa3 (Fig. 1a(i)). Each subunit contains at least one oligonucleotide/oligosaccharide-binding fold (OB), also known as the DNA-binding domain (DBD). Specifically, Rfa1 has four OB folds (DBD-F, DBD-A, DBD-B, and DBD-C), Rfa2 has one OB fold (DBD-D), and Rfa3 has one OB fold (DBD-E)[1,2]. Additionally, Rfa2 contains a winged helix–turn–helix (WH) domain at its C-terminus[7,8]. According to the current understanding, the WH domain and DBD-F are not involved

[1]Center for Quantitative Biology, Peking-Tsinghua Center for Life Sciences, Academy for Advanced Interdisciplinary Studies, Peking University, Beijing, China. [2]Department of Computational Medicine, University of California, Los Angeles, CA, USA. [3]Department of Molecular and Cellular Biochemistry, Indiana University, Bloomington, IN, USA. [4]BNLMS, College of Chemistry and Molecular Engineering, Peking University, Beijing, China. [5]Present address: Institute of Systems Biomedicine, School of Basic Medical Sciences, Peking University Health Science Center, Beijing, China. [6]These authors contributed equally: Jiawei Ding, Xiangting Li, Jiangchuan Shen. ✉e-mail: hniu@indiana.edu; zhiqi7@pku.edu.cn

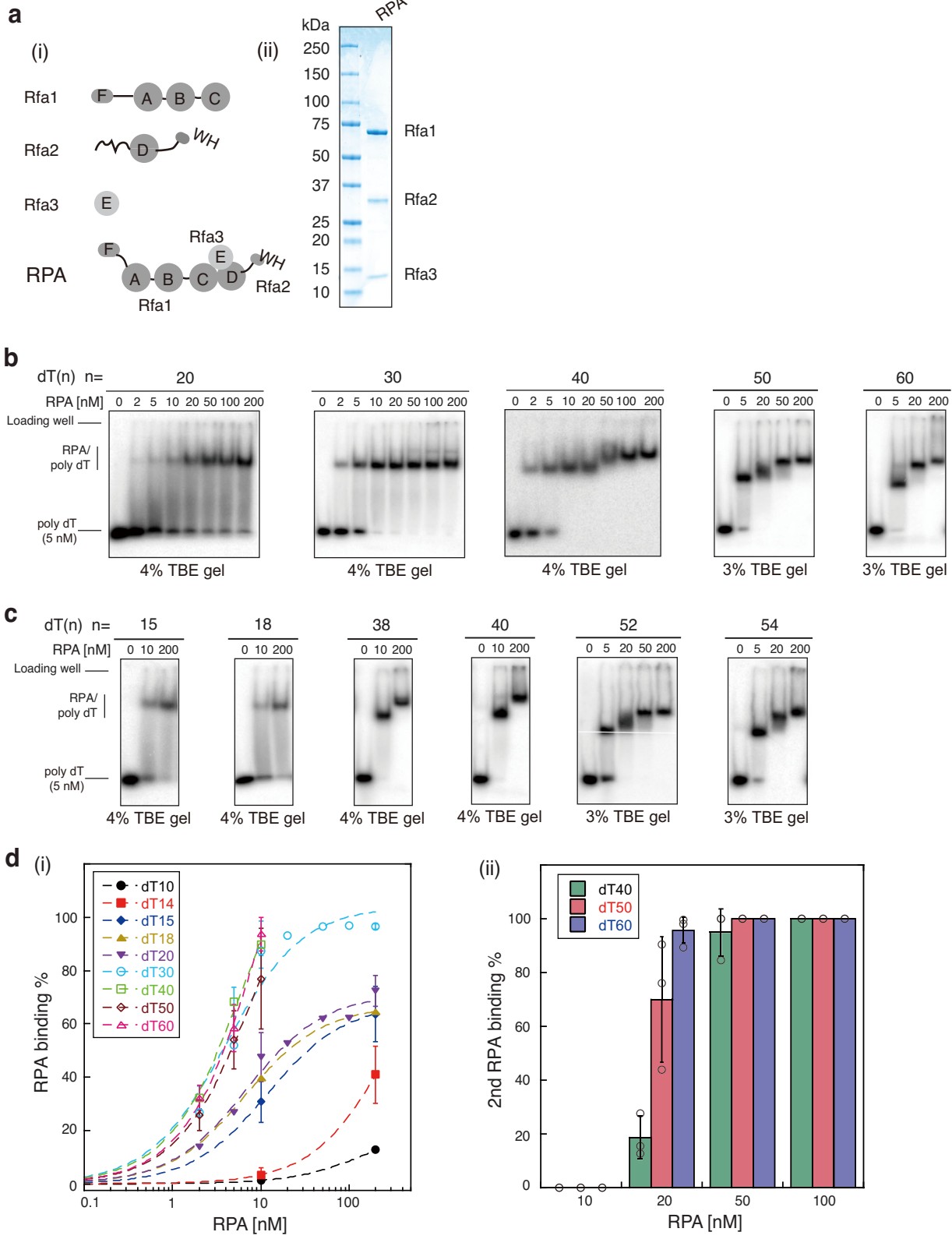

**Fig. 1 | EMSAs of Replication Protein A (RPA) binding to poly-dT ssDNA substrates with various length suggest the existence of multiple binding modes.**
**a** Schematic representation of Replication Protein A (RPA) from *Saccharomyces cerevisiae* (i) and purified RPA was analyzed on a 4–15% gradient SDS-PAGE (ii) and the experiment was repeated three times. **b** Titration of RPA (0–200 nM) to 5 nM 5′-labeled $dT_{(20)}$ ssDNA (i), $dT_{(30)}$ ssDNA (ii), $dT_{(40)}$ ssDNA (iii), $dT_{(50)}$ ssDNA (iv) and $dT_{(60)}$ ssDNA (v), at 150 mM KCl and the experiments were repeated three times.

**c** Titration of RPA (0-200 nM) to 5 nM 5′-labeled $dT_{(15)}$ ssDNA (i), $dT_{(18)}$ ssDNA (ii), $dT_{(38)}$ ssDNA (iii), $dT_{(40)}$ ssDNA (iv), $dT_{(52)}$ ssDNA (v) and $dT_{(54)}$ ssDNA (vi), at 150 mM KCl and the experiments were repeated three times. **d** Quantification of RPA binding percent (i) and 2nd RPA binding percent (ii) on the titration of RPA to key ssDNA substrates. Data are presented as mean ± SD in d. $N = 3$ independent experiments for each experimental condition in **d**. Source data are provided as a Source Data file.

in ssDNA binding; rather, they participate mainly in protein–protein interactions[2].

Despite weak individual affinity with ssDNA, the DBDs of RPA are able to tightly bind ssDNA through cooperative action with up to sub-nanomolar level dissociation constants. This cooperative behavior allows RPA binding to ssDNA dynamically through various binding modes[9,10]. The more stable modes involve occupying either 18–20 nucleotide (nt) (20-nt mode) or 28–30 nt (30-nt mode)[11]. The 30-nt mode, which involves binding of all four functional DBDs (A, B, C, and D), is considered the full-length binding mode, and has been previously reported to reside in a U-shape conformation[12].

The dynamic binding of RPA to ssDNA has been proposed to be crucial for its biological functions[3,13]. While RPA offers high affinity and protection of ssDNA, the shifting of RPA ssDNA binding modes likely frees ssDNA for downstream protein factors to grab ssDNA and act upon. In vitro, RPA ssDNA binding modes can be controlled by changes in biochemical and biophysical parameters. First, biochemical[14,15] and single-molecule fluorescence resonance energy transfer (smFRET) experiments[16] have both found that the concentration of RPA affects its ssDNA binding modes with higher concentration of RPA driving the transition from the 30-nt mode to the 20-nt mode or other partial binding modes. Second, both yeast and human RPA have been found to have their ssDNA binding modes affected by salt concentration[10,17]. Under the low salt conditions (20 mM NaCl), RPA prefers the 20-nt mode, whereas increasing the salt concentration drives the transition from the 20-nt to the 30-nt mode[17]. Interestingly, two different dissociation rates can be simultaneously detected when RPA molecules dissociate from a 35-dT ssDNA, strongly suggests the coexistence of two distinct RPA binding modes in the ssDNA-RPA complex[16,18].

In cells, RPA ssDNA binding modes are likely modulated by RIPs, such as Rad52[13] and Rtt105[19] in yeast. However, the mechanism behind this modulation is still unclear. In yeast, Rad52 is essential for HR mediated DSB repair, by acting as a mediator protein that assists the loading of Rad51 recombinase onto RPA coated ssDNA. In humans, this mediator function is largely fulfilled by the BRCA2 protein, mutations of which increase the risk of breast and ovarian cancer. Rad51 monomer binds weakly to ssDNA, however, once six or more Rad51 monomers polymerize on ssDNA, they are able to form stable nucleation filaments[20], that allow for further Rad51 polymerization and RPA displacement. While high concentration of Rad51 allow for RPA displacement in vitro, mediator proteins are indispensable for the Rad51 filament nucleation in cells. To fulfill its role as a mediator, the yeast Rad52 protein contains three functional domains: the N-terminal region (N domain) interacts with DNA and allows Rad52 to oligomerize; the middle region (M domain) interacts with RPA; and the C-terminal region (C domain) interacts with Rad51. RPA recruits Rad52 to the DNA damage site presumably due to their direct interaction and Rad52 further recruits and stabilizes Rad51 on ssDNA to aid in filament nucleation. Each Rad51 monomer occupies 3-nt ssDNA, requiring an 18-nt ssDNA footprint for filament nucleation. This space cannot be simply provided by the binding mode shift of a single RPA molecule, hinting the involvement of bimolecular dynamics of RPA. Moreover, many cellular processes generate ssDNA substrates of hundreds of nucleotides or even longer, such as DNA end resection in HR[21,22]. The long ssDNA substrates can host a large number of RPA molecules, whose stochastic behavior, albeit composing a complicated scenario, needs to be also considered to understand the control of ssDNA exposure.

While recent studies started to analyze the assembly of RPA molecules on long ssDNA substrates[16,23–25], the biophysical nature of how multiple RPA molecules bind to long ssDNA has not been carefully explored. This is mainly due to the lack of a system that can quantitatively measure how multiple RPA molecules can change their binding modes during dynamic binding to long ssDNA. In this study, we developed a three-step low-complexity ssDNA Curtains method, based

on single-molecule biophysics technology—ssDNA curtains[26,27]. By combining this single-molecule method and biochemical assays with a Markov chain model in non-equilibrium physics, we were able to quantitatively probe the dynamic binding behaviors of multiple RPA molecules on long ssDNA substrates. With this system, we identified the length extension state of ssDNA–RPA complex as an experimental readout to distinguish different RPA binding modes, and we further discovered that the ssDNA binding modes of RPA can be tuned by the Rfa2 WH domain and the RIPs, such as Rad52, to control the ssDNA accessibility. These results suggest a molecular mechanism for the mediator proteins and potentially other RIPs in action with RPA.

## Results

### EMSAs suggest RPA dynamic binding to ssDNA

To probe the binding behavior of multiple RPA molecules to ssDNA, we purified RPA using previously described methods[9,28] (Fig. 1a). Electrophoretic mobility shift assays (EMSAs) were conducted under the physiological salt concentration (150 mM KCl) to examine RPA binding to ssDNA substrates ranging from 20 nts to 60 nts with 10-nt steps (Fig. 1b). Titration of RPA from 2 to 200 nM with dT-20 and dT-30 resulted in a single major shifted species with similar affinities, confirming the stability of the 20-nt and 30-nt modes. Increasing the length of ssDNA substrate to 40 nts allowed stable binding of a second RPA molecule when the concentration of RPA reached 50–100 nM. In this scenario, both RPA molecules likely reside in the 20-nt mode on dT-40. While dT-50 substrate maintained the 2-stage binding pattern, on dT-60, the binding of the second RPA molecule was observed with 20 nM RPA, and the binding species of the third RPA molecule was clearly detected when RPA concentration reached 200 nM. The differences in allowable concentrations of secondary RPA occupancy on dT-40 and dT-60 substrates, along with the observation of the third RPA molecule's binding event, well indicated that while 30-nt mode is preferred when RPA concentration stays low and comparable to ssDNA substrate concentration, elevated RPA concentration is able to induce mode change from 30-nt mode to 20-nt mode[16]. Further fine size titration using oligonucleotide dT ssDNA substrates ranging from 10 nts to 60 nts revealed 15 nts, 40 nts and 54 nts being the minimal length for the stable binding of first, second, and third RPA, respectively (Fig. 1c, d, and Supplementary Fig. 1).

### Three-step low-complexity ssDNA Curtains approach to analyze RPA binding dynamics

Due to the low $K_D$ of RPA binding to ssDNA, bulk assays only allow us to examine RPA loading to ssDNA at equilibrium stage with near-saturated RPA concentration. In cells, however, following instantaneous long ssDNA formation during DNA replication and repair, most cases of RPA loading may occur at non-equilibrium stages or under locally non-saturated RPA concentrations. To quantitatively study RPA dynamics under such conditions in real time, we conducted ssDNA Curtains analyses[26], which have been used to study RPA and RIPs[9,19,29–31]. Unlike dsDNA, ssDNA coils up, thus the binding of fluorescently labeled RPA to ssDNA leads to an increase in both the length and the fluorescent intensity of the ssDNA-RPA complex. The fluorescent intensity is a precise measure of the amount of RPA molecules bound to ssDNA, which has been applied to analyze the binding and dissociation of RPA on ssDNA[9,29,31]. The length change of the ssDNA-RPA complex may be affected by both the number of RPA bound and the formation of DNA secondary structures. Additionally, the bending of ssDNA occurs when RPA fully occupies ssDNA in the 30-nt mode, which has been proposed to form a more compact structure[2,12]. Hence, the information of ssDNA binding mode change may be embedded in the length change of the ssDNA-RPA complex. We thus attempted to combine the analyses of both the intensity and length change of the ssDNA-RPA complex to quantitatively reveal the dynamics of RPA mode shift on long ssDNA.

To factor out the impact of DNA secondary structure formation on the length of ssDNA-RPA complex, we adopted the low-complexity ssDNA Curtains method[27], where a 29-nt long single stranded mini-circle with a TG-rich sequence was used as the template in rolling-circle reactions (RCRs) instead of the conventional M13mp18 phage genome[26,32] (Fig. 2a)[33]. To create RPA-MeGFP, a GFP tag with a single mutation (A206K) was introduced to hinder GFP dimerization without compromising RPA activities[9,28] or its function in cells[34]. Purified RPA-MeGFP (Supplementary Fig. 2a) binds 30-dT ssDNA similarly to RPA both in EMSA assays (Supplementary Fig. 2b and Fig. 1b(ii)) and in MicroScale Thermophoresis (MST) assays (Supplementary Fig. 2c, d), affirming the functionality of RPA-MeGFP in vitro.

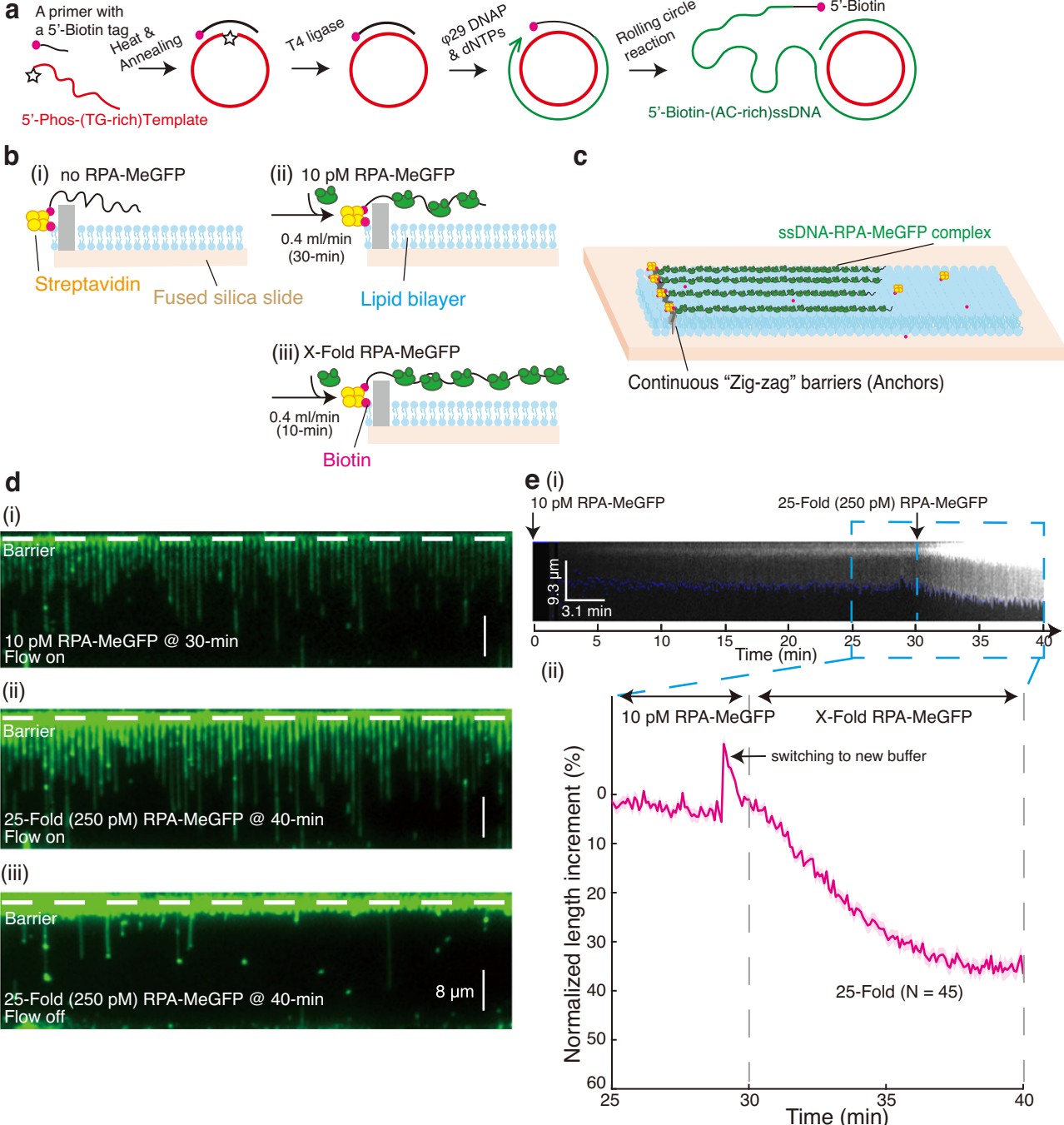

**Fig. 2 | Three-step low-complexity ssDNA Curtains. a** Outline of procedure for preparation of 5′ biotinylated ssDNA substrates with a low-complexity sequence by rolling circle replication of a designed ssDNA template. The template was 5′-/Phos/-TGG GTG TGT GTG TGT GTG TGT GTG GTG GT-3′. The biotinylated primer was 5′-/Biotin/-CAC CCA ACC ACC-3′. **b** Schematic of procedure for three-step low-complexity ssDNA Curtains. **c** Schematic of ssDNA Curtains. **d** Wide-field TIRFM images of ssDNA Curtains at 30-min time point with 0.4 mL/min flow of working buffer containing 10 pM RPA-MeGFP (i), or at 40-min time point with 25-fold RPA-MeGFP (250 pM) with flow on (ii) or off (iii). The green signals from RPA moved back with ssDNA molecules to the barrier when turning off the flow, which confirmed the specific binding of RPA to the ssDNA substrates. **e** (i) A representative kymograph showing the length dynamics of ssDNA–RPA complex from **d**. Blue dot line labeled the end tracking positions of a represented ssDNA–RPA complex, and the length analysis of the collective data of many ssDNA molecules was in (ii). N = 45, which was the total trace number of ssDNA molecules end tracking examined over three times DNA Curtains experiments. Data are presented as mean ± SEM in **e**. The working buffer was 40 mM Tris-HCl (pH 7.5), 2 mM MgCl₂, 0.2 mg/mL BSA, and 150 mM NaCl. Source data are provided as a Source Data file.

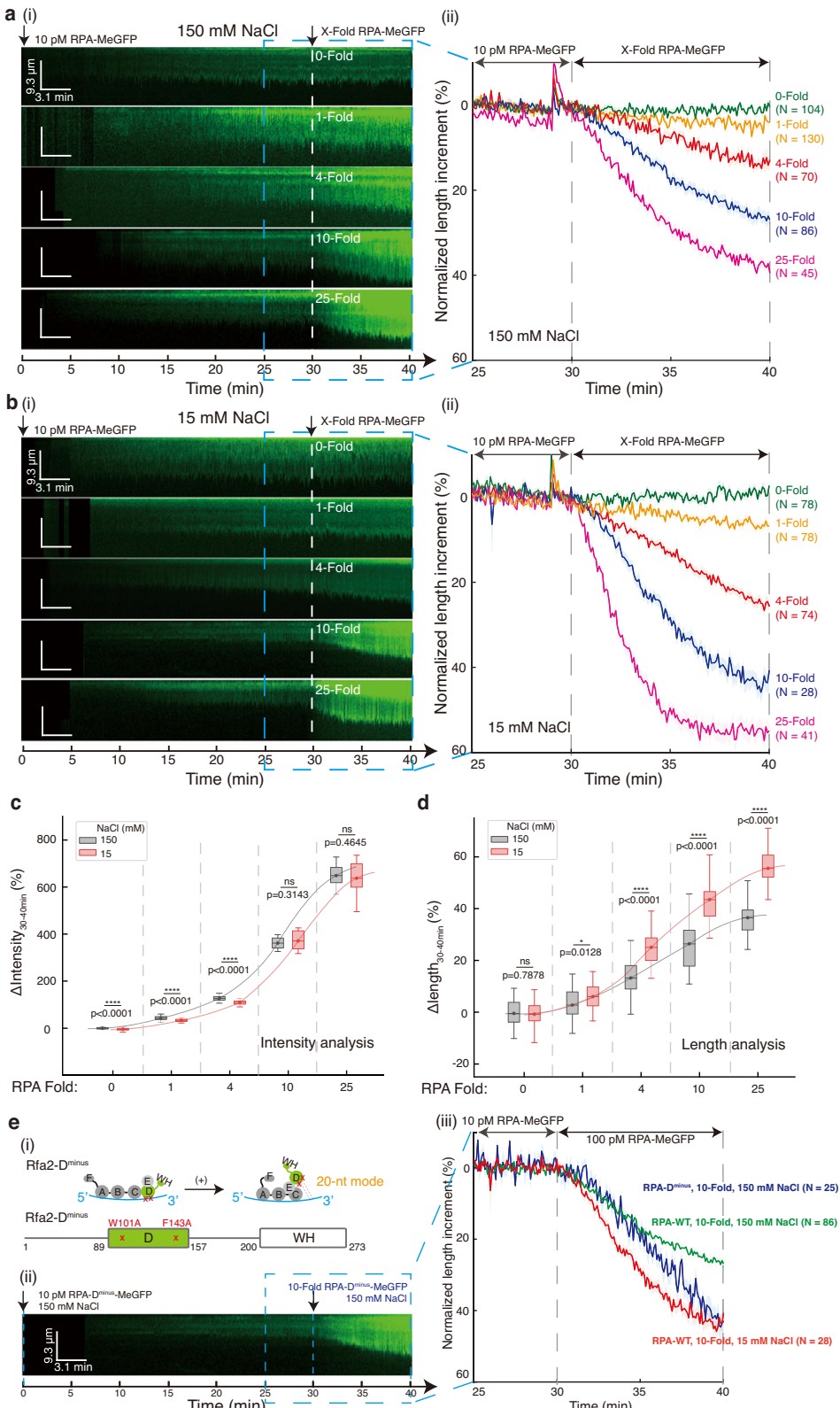

With these tools in hand, a three-step experimental procedure was further designed to adapt the length tracking analysis of the ssDNA–RPA complex (Fig. 2b). First, the low-complexity ssDNA substrates were injected into a microfluidic chamber, and these substrates were tethered onto individual lipids among the supported lipid bilayer within the chamber by Streptavidin-Biotin interactions (Fig. 2b(i), c). Second, the ssDNA substrates were pretreated with 10 pM RPA-MeGFP

for 30 min at 0.4 mL/min (Fig. 2b(ii)), a concentration that is 20-fold below its $K_D$ to ssDNA[17,35] and allows ssDNA substrates detection under a non-saturated RPA concentration to aid ssDNA length tracking analysis. Third, after the ssDNA-RPA complexes had reached a similar initiation state via the 30-min pre-treatment, the ssDNA substrates were treated with additional RPA-MeGFP at a customized concentration and under various buffer conditions (e.g. 15 mM versus 150 mM

**Fig. 3 | Salt and RPA concentrations affect RPA dynamic binding modes on long ssDNA. a, b** A set of experiments of the three-step low-complexity ssDNA Curtains. 150 mM NaCl in **a** and 15 mM NaCl in **b**. (i) Representative kymographs of 0, 1, 4, 10, 25-fold RPA; (ii) the length analysis of ssDNA-RPA complexes. **c, d** Boxplot of extension length change proportion in **c** or RPA-MeGFP intensity change proportion in **d** within the time window from 30-min to 40 min in **a, b**. Exact *p*-value (from left to right): 4.8e-5, 7.3e-8, 1.9e-9, 0.3143, 0.4645, 0.7878, 0.0128, 5.8e-10, 9.0e-8, 9.4e-12. Boxplot style: middle line (mean), box range (0.25–0.75), whisker range (min-max), with outliers removed. Statistics: one-way ANOVA (analysis of variation). *P*-value style: GP: ≥ 0.05 (ns), <0.05 (*), < 0.01 (**), < 0.001 (***), < 0.0001 (****).

**e** Binding dynamics of RPA-D$^{minus}$. (i) Schematic of RPA-D$^{minus}$ mutant, which attenuates DBD-D ssDNA binding affinity, was obtained by mutating Rfa2-W101A/ F143A. Representative kymographs of 10-fold RPA-D$^{minus}$-MeGFP with 150 mM NaCl were shown in (ii), and its length analysis (blue) was in (iii), where length dynamic curve with 10-fold RPA-WT at 150 mM NaCl (green) and 15 mM NaCl (red) condition from **a, b** were added for comparison. *N* represents the total trace number of ssDNA−RPA complex end tracking examined over three times DNA Curtains experiments for each experimental condition in **a, b,** and **e**. Data are presented as mean ± SEM in **a, b,** and **e**. *N* for each condition in **c, d** was same with **a, b**. Source data are provided as a Source Data file.

NaCl) for an additional 10 min at the same flow rate (Fig. 2b(iii)). The representative wide-field total internal reflection fluorescence microscope (TIRFM) images at the beginning and the end of the third step (30-min and 40-min time points, respectively) are shown in Fig. 2d and Supplementary Movie 1. By a signal tracking analysis of the ssDNA end position (marked by blue dots in Fig. 2e(i) and Methods), the length of a single ssDNA−RPA complex $L_t$ was measured as a function of time. Due to the heterogeneity in length of the RCR products, normalized length increment (Fig. 2e(ii)) was used instead of the absolute length as $(L_t - L_{30\,min})/L_{30\,min}$, where $L_{30\,min}$ was used as a reference length. When these ssDNA substrates, after 30-min RPA pretreatment, were exposed to RPA-free buffer (0-Fold RPA) for 10 min, the intensity of RPA on ssDNA remained stable under 150 mM NaCl ($\Delta int_{30-40\,min,\,150mM\,NaCl} = -0.1\% \pm 5.9\%$) and only slightly decreased under 15 mM NaCl ($\Delta int_{30-40\,min,\,15mM\,NaCl} = -4.8\% \pm 9.2\%, p < 0.0001$) (Supplementary Fig. 3a, b), which confirmed the stable association of RPA with ssDNA under 150 mM NaCl with a slightly reduced affinity under 15 mM NaCl. For comparison, under 150 mM NaCl, when RPA concentration increases 25-fold to 250 pM, a concentration close to the $K_D$ of RPA to ssDNA, the intensity of ssDNA-complex was increased by 648.3% ± 60.1% (Supplementary Fig. 3a), and the length of ssDNA−complex was increased by 36.5% ± 10.1% ($N = 45$, Fig. 2e(ii)). In contrast, with M13 ssDNA substrates (Supplementary Fig. 2c), ~80% compaction was observed within the first 30-min compared to our low complexity ssDNA substrates, which was further extended 359.2% ± 141.3% after switching to 100 pM RPA-MeGFP for another 10-min, likely due to the melting of extensive secondary structures by RPA. Hence, comparing to the traditional ssDNA Curtains approach, this three-step experimental design provided a synchronized initial state of the ssDNA−RPA complex to monitor RPA dynamics on long ssDNA substrates starting from a non-equilibrium state.

## Salt and RPA concentrations affect RPA binding modes on long ssDNA

In previous occlusion analysis, RPA predominantly adopted the 20-nt mode under low salt conditions, which shifts towards the 30-nt mode with increasing salt concentration[17]. Using the three-step experimental design, we conducted a set of low-complexity ssDNA Curtains for 1, 4, 10, and 25-Fold RPA (10, 40, 100, and 250, pM respectively) under both 150 mM NaCl and 15 mM NaCl conditions (Fig. 3a–d). In these assays, both the length and intensity were analyzed, where, for comparison, four characteristic variables were defined, viz. the initial rate of length ($v_{30\,min,len}$) or intensity ($v_{30\,min,int}$) increment at 30-min (Supplementary Fig. 3a–d) and the final length change ($\Delta len_{30-40\,min}$) or intensity change ($\Delta int_{30-40\,min}$) of ssDNA-RPA complex at 40-min (Fig. 3c, d, all values can be found in Supplementary Data 1).

The comparison of $\Delta int_{30-40\,min}$ between 15 mM NaCl and 150 mM NaCl conditions (Fig. 3d) demonstrated that a slightly reduced number of RPA molecules was loaded onto ssDNA substrates under 15 mM NaCl condition with 1-Fold ($p < 0.0001$) and 4-Fold ($p < 0.0001$) RPA. Increasing the buffer RPA concentration minimized the difference in the number of RPA molecules finally loaded ($p = 0.3143$ (n.s.) with 10-fold RPA and $p = 0.5336$ (n.s.) with 25-fold RPA) (Fig. 3d). In EMSA, binding of RPA (2 to 200 nM) to short ssDNA substrates under 15 mM

salt was similar to RPA under 150 mM salt (Supplementary Fig. 1b). However, we found that low salt, while barely affecting the amount of RPA loaded, significantly extended the length of the ssDNA−RPA complex. The analysis of $\Delta len_{30-40\,min}$ (Fig. 3a–c) indicated that the final extension length of ssDNA-RPA complexes at 15 mM NaCl were 3.2% ± 2.5%, 11.8% ± 1.0%, 17.1% ± 3.3%, 19.1% ± 4.2% longer than at 150 mM NaCl with 1-Fold, 4-Fold, 10-Fold, and 25-Fold RPA, respectively, all with significant differences. As a control, changing the salt concentration of the 30-min preincubation to 15 mM instead had no impact on RPA binding at the third step of 25-fold RPA under 150 mM NaCl (Supplementary Fig. 3e). Thus, the impact of the preloaded RPA status on the final RPA binding was negligible.

We suspect that the differences in ssDNA−RPA length extension observed under 15 mM and 150 mM NaCl may be caused by the shift between the RPA 20-nt ssDNA binding mode and the 30-nt mode. To test this premise, knowing that the major difference between the 20-nt and the 30-nt mode is the occupancy of the DBD-D domain on ssDNA, we mutated the key aromatic residues in the DBD-D involved in ssDNA binding to obtain DBD-D$^{minus}$, which attenuates its association with ssDNA. Consistent with previous studies[35], the resulting RPA-D$^{minus}$ did not significantly altered the binding of RPA to short ssDNA substrates in EMSA (Supplementary Fig. 4a(i), b). In ssDNA Curtains, under 150 mM NaCl condition, RPA-D$^{minus}$ (Fig. 3e) at 10-fold concentration had slightly lower number of RPA molecules loaded ($p = 0.0002$), but extended the ssDNA−RPA complex significantly longer than the RPA-WT ($p < 0.0001$) and was comparable to RPA-WT under 15 mM NaCl condition by the 40-min end point of recording ($p = 0.9123$ (n.s.)). Therefore, there is likely a positive correlation between the 20-nt mode of RPA occupancy and the longer extension state of the ssDNA-RPA complex on long ssDNA substrates.

Interestingly, RPA-D$^{minus}$ under 150 mM NaCl condition showed a similar extension length comparing to RPA-WT under 15 mM NaCl condition, albeit with a slower kinetics (Fig. 3e(iii)). Increasing the concentration of RPA-D$^{minus}$ and RPA-WT to 25-Fold (Supplementary Fig. 4c) reduced the difference between their extension kinetics. Moreover, with 25-fold RPA-D$^{minus}$, a ssDNA−RPA complex with even longer extension comparing to RPA-WT was observed at the 40-min end point likely due to the enhanced RPA binding at higher concentrations (Supplementary Fig. 4d). Our ssDNA Curtains experiments again confirmed that the ssDNA bound by RPA in 20-nt mode forms a longer extended conformation in contrast to RPA in the 30-nt mode, which provided an experimental readout to monitor the dynamic shifting of RPA binding modes. Altogether, these results suggested that the RPA 20-nt ssDNA binding mode is preferred under 15 mM NaCl and the preference is shifted to the 30-nt mode upon increasing salt concentration.

## A stochastic model for multiple RPA molecules binding to long ssDNA

Our three-step low-complexity ssDNA curtains experiments provide information on the overall dynamic behaviors of multiple RPA molecules on long ssDNA from its length change and intensity change. However, these experiments alone cannot provide the exact molecular ratio or the specific position distribution of the 20-nt mode and the

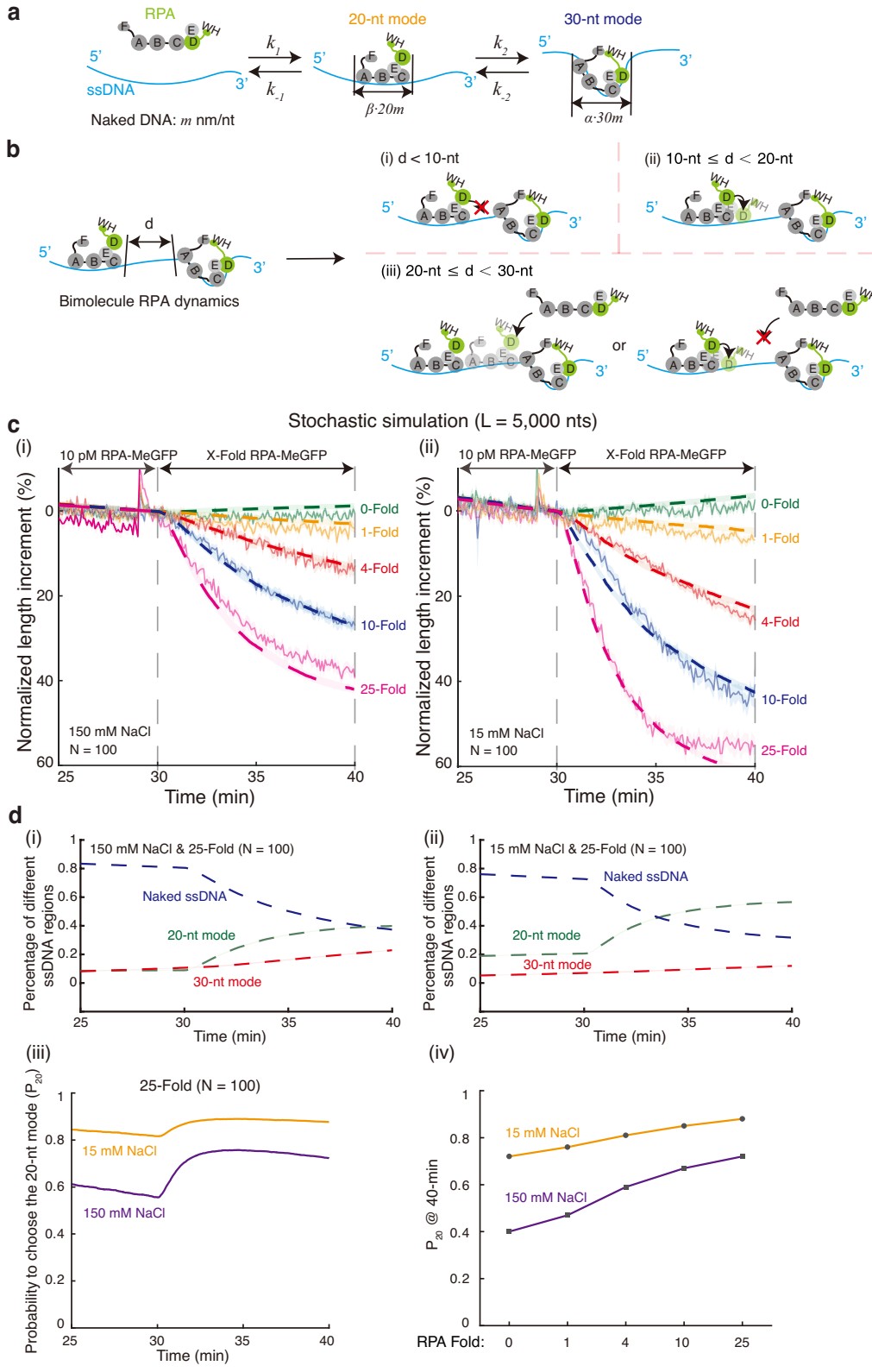

**c** Stochastic simulation (L = 5,000 nts)

30-nt mode RPA on long ssDNA, or the information on the spacing between neighboring RPA molecules. We next investigated whether we could establish a theoretical model to decipher the dynamic mechanism of multiple RPA molecules binding to long ssDNA (Fig. 4 and Supplementary Information). One-dimensional random sequential adsorption (1D-RSA) model[36] provided an ideal basis for further modelling the processes of multiple protein binding to ssDNA.

Relevant theoretical consequences of applying RSA-like model to DNA–protein reaction kinetics are also known as the McGhee-von Hippel model[24,37–39]. Thus, we established a continuous-time discrete Markov chain model for multiple RPA molecules dynamic binding to long ssDNA.

We simplified RPA binding modes as a 20-nt mode and a 30-nt mode, respectively representing a partial binding mode (PBM) and a

**Fig. 4 | A continuous-time discrete Markov chain model for multiple RPA molecules binding to long ssDNA. a** Schematic of building up continuous-time discrete Markov chain model. $m$ represents absolute extension length of naked ssDNA per nt under a constant force in 5′–3′ direction. All parameters are explained in Supplementary Information in detail. **b** All possible behaviors of bimolecular dynamics of RPA on ssDNA, which composed a complicated scenario of multiple RPA molecules binding. $d$ represents the size of the naked ssDNA gap between two randomly loaded RPA molecules. **c** Stochastic simulations of the Markov chain model (dash lines) at 150 mM NaCl (i) and 15 mM NaCl (ii). These simulations fit well with experimental results (solid lines), which were also shown in Fig. 3a, b. Length of

ssDNA substrate ($L$) used in stochastic simulations was 5000 nts, and stochastic simulation for each condition was repeated for 100 times ($N = 100$). **d** Stochastic simulation for the 25-Fold condition. (i–ii) Real-time changes of ssDNA region percentage bound by RPA in 20-nt mode (green), 30-nt mode (red) and no RPA (blue) at 150 mM NaCl (i) and 15 mM NaCl (ii) were plotted. (iii–iv) The probability to choose the 20-nt mode ($P_{20}$) was plotted. $P_{20}$ was defined as the proportion of 20-nt mode RPA bound among total number of RPA bound. (iii) Real-time changes of $P_{20}$ at 150 mM NaCl and 15 mM NaCl were plotted. (iv) $P_{20}$ values at 40-min time point of 0, 1, 4, 10, 25-fold RPA at 150 mM NaCl and 15 mM NaCl were compared. Data are presented as mean ± SD in **c**, **d**. Source data are provided as a Source Data file.

full-length binding mode (FLBM). RPA binds to ssDNA with a polarity, and DBD-A binds at the 5′ end and DBD-D binds at the 3′ end. When an RPA binds to ssDNA, DBD-A, B, and C together firstly bind to ssDNA, forming the 20-nt mode[2]. The subsequent DBD-D binding leads to the 30-nt mode. To build up the model (Fig. 4a), RPA initially binds to a 20-nt ssDNA with a rate of $k_1$ (unit s⁻¹, the concentration of ssDNA and RPA incorporated), and dissociates from ssDNA with a rate of $k_{-1}$ (unit s⁻¹). Afterwards, the DBD-D starts to bind to an extra 10-nt ssDNA with a rate of $k_2$ (unit s⁻¹, the concentration of ssDNA and RPA incorporated), and dissociates from ssDNA with a rate of $k_{-2}$ (unit s⁻¹).

By using 5000 nts as the length of ssDNA and considering volume exclusion effects, we conducted stochastic simulations ($N = 100$) to probe all possible behaviors of multiple RPA molecules binding to long ssDNA, which can be distilled to bimolecular RPA dynamics (Fig. 4b). Simulations returned the trajectories of DNA occupation fractions by RPA in the 20-nt binding mode or in the 30-nt mode respectively. We introduced additional length scaling parameters $\alpha$ and $\beta$ to convert the occupation fractions to the modulated length trajectories. $\alpha$ and $\beta$ respectively represent the relative unit extension of DNA bound by RPA in the 20-nt mode and in 30-nt mode with respect to unit length m (unit nm/nt) of the naked ssDNA. The parameters used for simulation are determined through an iterative procedure to update parameters and minimize the associated least squared error (Supplementary Fig. 5a–f). The results of stochastic simulations for the 150 mM NaCl condition and 15 mM NaCl condition are shown in Fig. 4c(i)–(ii), and the simulation parameters are shown in Supplementary Fig. 5g.

The stochastic simulations can fit the experimental data well (Fig. 4c). With obtaining the exact molar ratio and specific spatial distribution of the 20-nt mode and the 30-nt mode RPA on long ssDNA, the effect of low salt and high amount of loaded RPA in modulating RPA binding modes got further testified by our model. From the model, we were able to plot DNA occupation fractions by RPA in the 20-nt mode, the 30-nt mode, or no RPA respectively during the entire 40-min reaction process as seen in Fig. 4d(i)–(ii) and Supplementary Fig. 6a. We can use this information to calculate how many RPA molecules opt for the 20-nt mode or the 30-nt mode as well as the length of naked ssDNA on ssDNA-RPA complex across each time point. The calculated results at 40-min time point are listed in Supplementary Fig. 6b.

To demonstrate the effect of salt and loaded RPA number on the shift between RPA binding modes, we used 25-Fold as an example (Fig. 4d). We defined $P_{20}$ as the probability of bound RPA to choose the 20-nt ssDNA binding mode, which was calculated as the number of RPA in the 20-nt mode over the entirety of loaded RPA onto ssDNA. In comparison to the 150 mM NaCl condition, RPA at the 15 mM NaCl condition had a significantly larger value of $P_{20}$ (Fig. 4d(iii)–(iv)), implicating that RPA at low salt condition prefers the 20-nt mode. As RPA molecules gradually loaded onto the proportions of naked ssDNA (Fig. 4d(iii)) or with increasing RPA concentration (Fig. 4d(iv)), the value of $P_{20}$ also increased correspondingly, suggesting more loaded RPA on ssDNA or higher RPA concentration can stimulate an increased number of RPA molecules to shift to the 20-nt mode.

These findings acquired from the model perfectly match the effect of low salt and high amount of RPA on inducing the 20-nt mode

concluded from the ssDNA Curtains analyses. The exact number of transient molar ratio and position distribution at any specific time point can be obtained from the stochastic model. Our data suggest that the 20-nt mode is the predominant RPA binding mode under high RPA concentration, especially at the 15 mM NaCl condition, while the 30-nt mode is the predominant RPA binding mode with low RPA concentration and is more preferred at 150 mM NaCl condition. When more RPA loads and accumulates onto ssDNA, the 20-nt mode begins to be preferred over the 30-nt mode.

## ssDNA gaps in between RPA molecules aid Rad51 assembly onto RPA-coated ssDNA

In HR pathway, following 3′-ssDNA exposures by DNA end resection, RPA occupies ssDNA and removes DNA secondary structures in order for Rad51 to be loaded. However, RPA, possessing a ssDNA binding affinity more than 1000 fold higher than a Rad51-ATP monomer ($K_D \sim 200$ nM)[40], is a strong competitor of the RecA/Rad51 family of recombinases on ssDNA. Nonetheless, polymerization of Rad51 on ssDNA, a process aided by mediator proteins, e.g., Rad52 in yeast and BRCA2 in humans, stabilizes Rad51 on ssDNA and allow it to displace RPA. The key step in this process is the allowance of six Rad51 monomers each with a 3-nt footprint forming a stable nucleation cluster[41] that occupies a 18-nt long ssDNA region (Fig. 5a). High concentration Rad51 is capable to form dispersed nucleation spots on ssDNA–RPA complex in vitro[29,42], implicating the presence of naked ssDNA regions upon RPA binding. Real-time analyses on RPA position distribution and RPA spacing by our stochastic model suggest the presence of naked ssDNA regions between RPA molecules bound on ssDNA, which we call ssDNA gaps. For instance, in our simulations, after 10-min flush of 250 pM RPA under 150 mM NaCl condition, unoccupied ssDNA regions were reduced from ~80% to ~37% (Fig. 4d(i)–(ii)), but not to zero. Analysis of ssDNA gap size distribution demonstrated that most of the ssDNA gaps (~90%) formed under this condition were smaller than 30-nt (Supplementary Fig. 6c). Our model allowed us to further analyze the total number of ssDNA gaps with a size of 18-nt or larger at the 40-min time point, which decreases drastically upon increasing concentration of RPA in the system (Fig. 5b). To understand how the difference in ssDNA gap size distribution affects Rad51 loading, using our three-step low-complexity ssDNA Curtains design, we added a fourth step of 10 min stop-flow incubation of Rad51 in RPA-free buffer on ssDNA Curtains with a double-tethered pattern to maintain the extension state of ssDNA–RPA complexes under a stopped flow (Fig. 5c). Displacement of RPA-MeGFP by dark Rad51 leads to loss of fluorescent intensity. We thus analyzed the decrease in intensity throughout the 10-min Rad51 incubation to assess the RPA displacement efficiency by Rad51. While the occurrence frequency of ssDNA gaps (≥18-nt) under 150 mM NaCl condition with 10-Fold RPA is 1.98-fold higher than with 25-Fold RPA ($p < 0.0001$), we found that $60.1 \pm 10.0\%$ RPA molecules were finally displaced by Rad51 on the ssDNA-RPA complex treated with 10-Fold RPA. Only $11.9 \pm 11.0\%$ RPA were replaced with 25-Fold RPA ($p < 0.0001$) (Fig. 5d). Hence, naked ssDNA gaps likely exist between RPA molecules on the ssDNA–RPA complex, and the ssDNA-RPA

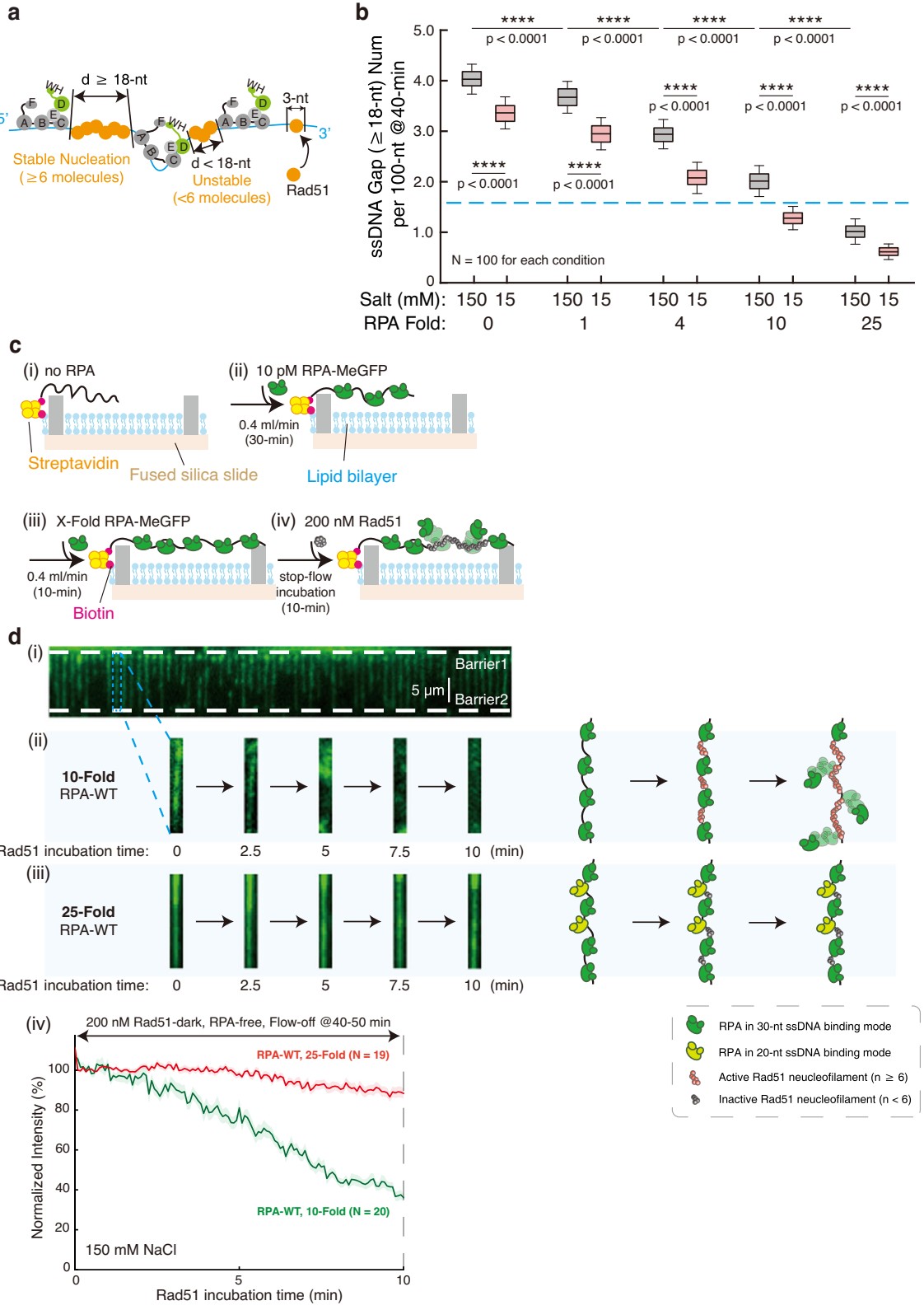

complex with more ssDNA gaps (≥18-nt) is more prone to Rad51 assembly.

## The Rfa2 WH domain regulates the RPA ssDNA binding mode and ssDNA accessibility

In our simulations, the occurrence frequency of ssDNA gaps (≥18-nt) on the ssDNA-RPA complex was significantly enhanced under 150 mM NaCl compared to 15 mM NaCl (Fig. 5b). We interpret this as meaning

that the 20-nt mode, which is preferred by RPA under 15 mM NaCl, may favor a tighter RPA packing and a smaller ssDNA gap size (Fig. 6a). Thus, the ssDNA gap size may be regulated by cells through modulating the changes between the 20-nt and the 30-nt mode of ssDNA binding by RPA. Besides the four DBDs, many RIP interactions were mediated by the Rfa1-N domain and the Rfa2-WH domain[2], which may regulate the RPA ssDNA binding modes. We therefore cloned and purified RPA-ΔWH (Fig. 6b(i) and Supplementary Fig. 4a(ii)). The

**Fig. 5 | Naked ssDNA gaps dynamically exist between RPA molecules and facilitates Rad51 assembly onto RPA-coated ssDNA. a** Schematic of Rad51 nucleofilament formation on ssDNA–RPA complex. **b** Boxplot of ssDNA gaps with size ≥18-nt on simulated ssDNA-RPA complex at 40-min time point for 0, 1, 4, 10, 25-fold RPA at 150 mM NaCl (gray) and 15 mM NaCl (red). Exact *p*-value (from left to right): 8.2e-58, 1.3e-26, 5.8e-61, 1.4e-64, 6.0e-77, 3.1e-82, 2.7e-72, 1.9e-92, 3.1e-50. *N* = 100 repeated simulations for each condition. Boxplot style: middle line (mean), box range (0.25-0.75), whisker range (min-max), with outliers removed. Statistics: one-way ANOVA (analysis of variation). *P*-value style: GP: ≥ 0.05 (ns), <0.05 (*), < 0.01 (**), < 0.001 (***), < 0.0001 (****). **c** Schematic of procedure for Rad51 nucleofilament formation based on the double-tethered ssDNA Curtains. **d** Rad51

formed nucleofilament on ssDNA–RPA complex and displaced RPA. Representative snapshots of ssDNA–RPA complex at 0, 2.5, 5, 7.5, 10 minutes after Rad51 injection were picked from the experimental results of the double-tethered ssDNA Curtains of the 10-Fold RPA-WT condition in (i) and the 25-Fold RPA-WT condition in (ii). The salt condition was 150 mM NaCl. (iii) Quantification of MeGFP intensity decrease during 10-min Rad51 incubation in (i) and (ii) in RPA-free working buffer. The 10-Fold RPA-WT condition, *N* = 20; The 25-Fold RPA-WT condition, *N* = 19. Data are presented as mean ± SEM in d. *N* represents the total trace number of ssDNA–RPA complex end tracking examined over three times DNA Curtains experiments for each experimental condition in **d**. Source data are provided as a Source Data file.

binding affinity of RPA-ΔWH to short ssDNA substrates in EMSAs was comparable to RPA-WT (Supplementary Fig. 7a). Nonetheless, in our three-step DNA Curtains analyses, RPA-ΔWH under 150 mM NaCl condition indeed mimicked RPA-WT under 15 mM NaCl condition both in length and intensity change (all n.s., as shown in Fig. 6b, c and Supplementary Fig. 7b, c). Reducing the salt concentration to 15 mM NaCl did not alter the length and intensity changes of RPA-ΔWH with 10-Fold and 25-Fold RPA (all n.s., as shown in Fig. 6b, c and Supplementary Fig. 7b, c), though with a slightly slower kinetics of 25-Fold RPA-ΔWH at 15 mM NaCl. Thus, inactivation of the Rfa2 WH domain triggers more RPA molecules to adopt the 20-nt mode on long ssDNA. To test this premise, we injected Rad51 after a 10-Fold RPA-ΔWH flush with 150 mM NaCl and found that the RPA displacement efficiency by Rad51 decreased significantly to 22.7% ± 11.0% from 60.1% ± 10.0% with 10-Fold RPA-WT (*p*<0.0001) (Fig. 6d(i)–(ii)). The RPA displacement efficiency by Rad51 within 10 min of the 10-Fold RPA-WT, 25-Fold RPA-WT and 10-Fold RPA-ΔWH exhibits a clear linear correlation ($r^2 = 0.9993$) with the occurrence frequency of ssDNA gaps (≥18-nt) (Fig. 6d(iii)). These results further confirm that Rad51 assembly on the ssDNA-RPA complex is realized through the spacing property of RPA loading, which leaves naked ssDNA gaps whose size can be tuned by regulating the RPA ssDNA binding modes.

### Rad52 promotes Rad51 nucleofilament formation by modulating RPA spacing on ssDNA

In yeast, Rad52 functions as the major mediator during HR to facilitate Rad51 assembly on the ssDNA-RPA complex. Rad52 contains three functional domains, the N-terminal domain mainly contributes to its oligomerization and DNA interaction, the C-terminal domain contributes to its Rad51 interaction, and the middle acidic domain (Rad52-M) contributes to its RPA interaction. RPA interaction targets Rad52 to the ssDNA-RPA complex in cells[43] and is required for both its mediator function[43] and its role in promoting single strand annealing in the presence of RPA[44]. The Rad52-M domain interacts with both Rfa1 and Rfa2-WH domain[2]. We next ask whether the physical interactions between the Rad52-M domain and RPA directly change the RPA ssDNA binding mode and RPA displacement by Rad51. We therefore cloned and purified the Rad52-M domain (Fig. 7a(i)) and examined it using our three-step ssDNA Curtains analyses by injecting RPA that was pre-incubated with the Rad52-M domain during the third step. Interestingly, the injection of RPA and Rad52-M resulted in a significantly shorter ssDNA–RPA complex at 150 mM NaCl with 10-Fold RPA (Fig. 7a, b) (*p* = 0.0128) with minimal change in the amount of RPA loaded (Fig. 7b) (*p* = 0.2332 (n.s.)), likely due to the impact of the Rad52-M domain in stabilizing RPA in its 30-nt mode. Interestingly, the addition of the Rad52-M domain nearly completely eliminated the extra extension of ssDNA–RPA complex observed with RPA-ΔWH (Fig. 7b(i) and Supplementary Fig. 7c) (*p* < 0.0001), with no apparent change in the loaded amount of RPA (Fig. 7b(ii) and Supplementary Fig. 7d) (*p* = 0.7673 (n.s.)), which indicates the presence of a second binding mode-controlling module on RPA besides the WH domain subjective to Rad52 regulation.

Since the RPA 30-nt ssDNA binding mode resulted in a larger ssDNA gap size, we next tested whether the Rad52-M domain had an effect on RPA displacement by Rad51. Surprisingly, the presence of the Rad52-M domain enhanced the RPA displacement efficiency of Rad51 by over three-fold, from 11.9 ± 11.0% to 35.9 ± 6.9% (*p*<0.0001) (Fig. 7e). Taken together, our results revealed that Rfa2-WH domains function as the module to control dynamic change of RPA between the 20-nt and the 30-nt ssDNA binding mode, which can be orchestrated by the Rad52 to fulfill its mediator function.

## Discussion

To summarize, we have developed a three-step low-complexity ssDNA Curtains platform and combined it with a Markov chain model in non-equilibrium physics to quantitatively examine the dynamic binding of RPA molecules on long ssDNA substrates. Applying the length analysis in ssDNA Curtains allowed us to capture transient changes in conformation of ssDNA–RPA complex which reflect changes in the RPA ssDNA binding modes. Our results suggest that the dynamic ssDNA exposure between neighboring RPA molecules can facilitate Rad51 nucleation on RPA coated ssDNA. The resulting ssDNA gaps rapidly change in size due to binding and unbinding of RPA domains but have a relative stable size distribution, which is controlled by the shifting between the partial binding modes (PBMs, e.g. 20-nt mode) and the 30-nt full-length binding mode (FLBM) of RPA ssDNA binding. Tighter spacing between RPA molecules is favored under the PBMs of RPA ssDNA binding, which can be regulated by the Rfa2 WH domain. In our results, RPA-ΔWH shifts RPA to the 20-nt mode and significantly inhibits Rad51 assembly on the ssDNA-RPA complex. Notably, in ssDNA Curtains, the Rad52-M domain, which only interacts with RPA but not with Rad51 or ssDNA, can adjust RPA spacing to promote Rad51 nucleofilament formation on RPA coated ssDNA.

The system we devised allowed us to monitor RPA loading from a non-equilibrium state. Despite the high affinity of RPA to ssDNA ($K_D$ ~ 200 pM), ssDNA-RPA can hardly reach equilibrium state where ssDNA is fully covered by RPA even when cumulatively ~1000-fold saturated RPA was provided through a 10-min 0.4 mL/min treatment of 250 pM RPA (Fig. 3a, b and Supplementary Fig. 3a, b). In consistence with this observation, simulations based on our stochastic model of RPA binding to long ssDNA demonstrated that even saturated RPA left ~20% ssDNA region dynamically exposed rather than tightly aligned (Fig. 4d). Williams and colleagues have used a similar model to study the dynamics of *Escherichia coli* SSB[24], where a standard mean field approximation is employed to obtain accurate analysis of the kinetic parameters. Here, the unique feature of our stochastic model is that an accurate stochastic sampling approach was implemented to reveal finer structures at nonequilibrium state such as the ssDNA gap size distribution (Supplementary Fig. 6c), which likely plays an important role in regulating Rad51 nucleation that demands a minimal 18-nt footprint. For one case among the complicated scenarios (Fig. 4b), a 15-nt ssDNA gap would only allow for 5 Rad51 monomers, but can easily satisfy Rad51 nucleofilament formation when the DBD-D domain of the 5′-RPA transiently dissociate from ssDNA. In the ssDNA Curtains analysis, exposed ssDNA

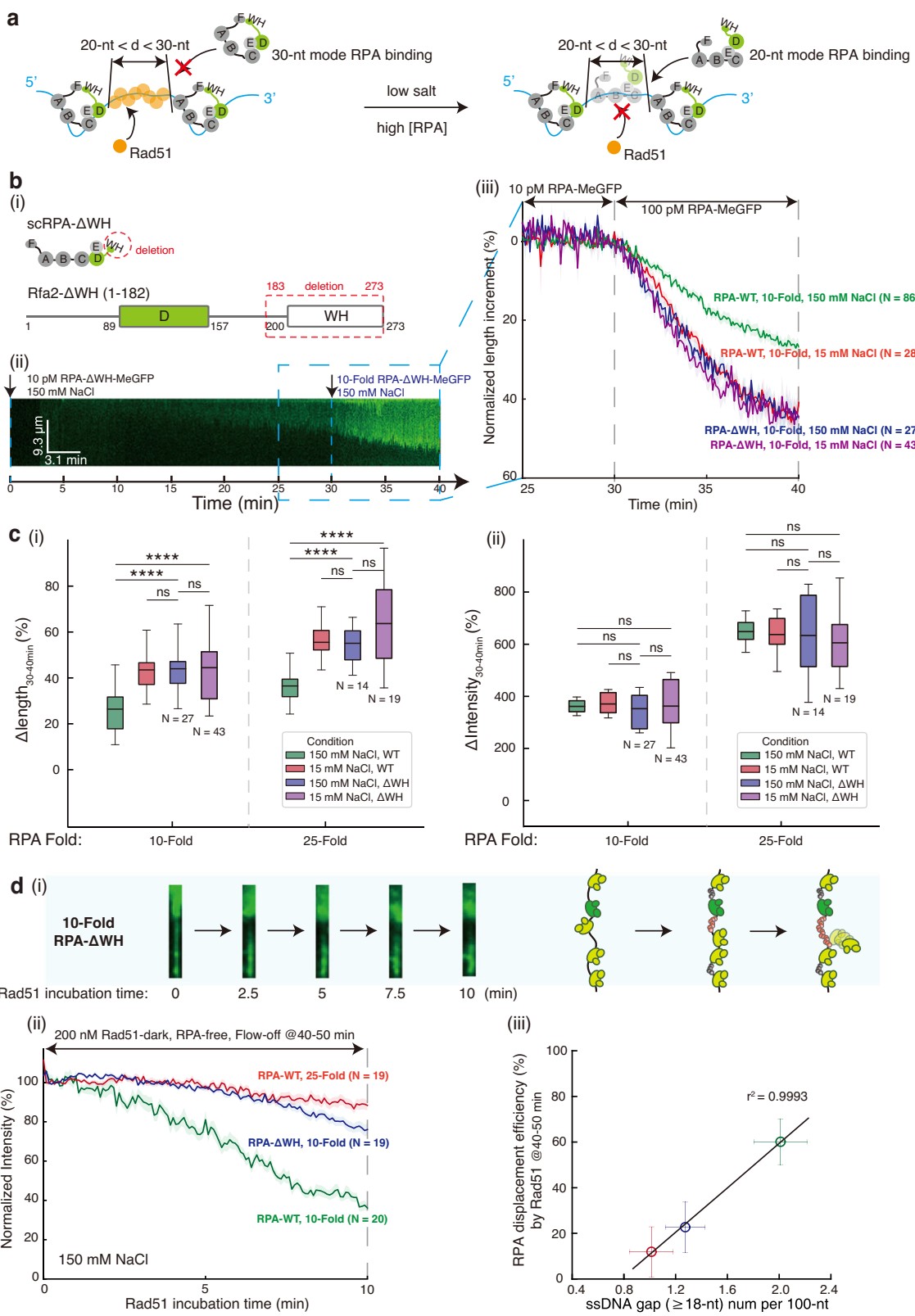

region on ssDNA–RPA complex formed by 10-min 0.4 mL/min treatment of 100 pM RPA enabled ~60% of the loaded RPA to be replaced by the following 10-min incubation with 200 nM Rad51 (Fig. 5d) in the absence of free RPA. Notably, when the ssDNA-RPA complexes in ssDNA Curtains experiments were switched into RPA-free buffer, the off rate of bound RPA was negligible (Fig. 3a–d), thus, the RPA displacement by Rad51 is likely not due to RPA turnover, but Rad51

polymerization following its nucleation. We did not include cooperativity into our RPA dynamic binding model because RPA has been previously proved to bind ssDNA with low cooperativity[14,17,45], which is also supported by our EMSAs and ssDNA Curtains data (Figs. 1 and 3). Recently, CryoEM and FRET studies have reported that S178 phosphorylation regulates scRPA binding to ssDNA by inducing cooperativity[23]. Impact of the potential cooperative binding under

**Fig. 6 | The Rfa2-WH domain regulates RPA ssDNA binding mode and ssDNA accessibility on ssDNA–RPA complex. a** Schematic of how RPA ssDNA binding modes affect RPA spacing and ssDNA accessibility on ssDNA-RPA complex. **b** Binding dynamics of RPA-ΔWH. (i) Schematic of RPA-ΔWH mutant. Representative kymographs of 10-fold RPA-ΔWH at 150 mM NaCl were shown in (ii), and its length analysis (blue) was in (iii), together with 10-fold RPA-ΔWH at 15 mM NaCl (purple). compared with RPA-WT (red and green) from Fig. 3a, b. **c** Boxplot of length (i) or intensity (ii) change proportion within 30–40-min of ssDNA-RPA complexes from 10-fold and 25-fold RPA-ΔWH at 150 mM NaCl (blue) and 15 mM NaCl (purple), compared with RPA-WT (red and green). Exact *p*-value (from left to right, from top to bottom): 2.4e-8, 3.0e-7, 0.8824, 0.9207, 1.2e-9, 3.3e-8, 0.9168, 0.1986; 0.9353, 0.5451, 0.4849, 0.7993, 0.1085, 0.6655, 0.5263, 0.6498. Boxplot style: middle line (mean), box range (0.25–0.75), whisker range (min-max), with outliers removed. Statistics: one-way ANOVA (analysis of variation). *P*-value style:

GP: ≥ 0.05 (ns), <0.05 (*), < 0.01 (**), < 0.001 (***), < 0.0001 (****). **d** RPA-ΔWH inhibits Rad51 nucleofilament formation on ssDNA–RPA complex. Representative snapshots of ssDNA-RPA complex at 0, 2.5, 5, 7.5, 10 min after Rad51 injection of 10-Fold RPA-ΔWH at 150 mM NaCl (i). Quantification of GFP intensity decrease during 10-min Rad51 incubation for 10-Fold RPA-ΔWH (blue) in RPA-free working buffer and comparison with RPA-WT (red and green) (ii). Correlation between RPA displacement efficiency by Rad51 during 10-min Rad51 incubation and ssDNA gap (≥ 18-nt) event number from Fig. 5b was plotted in (iii), where the three points were 10-Fold RPA-WT (green), 10-Fold RPA-ΔWH (blue) and 25-Fold RPA-WT (red). Data are presented as mean ± SEM in **b** and **d**(ii), and mean ± SD in **d**(iii). *N* represents the total trace number of ssDNA–RPA complex end tracking examined over three times DNA Curtains experiments for each experimental condition in **c**, **d**. Source data are provided as a Source Data file.

certain conditions on the RPA ssDNA binding modes would be an interesting research direction.

By connecting the RPA ssDNA binding modes transition with RPA spacing on the ssDNA–RPA complex, our study also revealed an interesting molecular mechanism for cellular regulation of ssDNA accessibility, where RPA shifting from the 30-nt to the 20-nt mode tightens spacing between RPA molecules (Fig. 8). Previous cross-linking results suggested that 20-nt mode is a stable PBM of RPA with engaging of DBD-A, B and C[35]. Our EMSAs of RPA binding to oligo-dT substrates also affirmed the 20-nt mode and the 30-nt mode being stable. Noted that out model only considered 20-nt and 30-nt mode, but not 10-nt mode, which requires a very high concentration of RPA though, but may provide a tighter spacing. The RPA-D$^{minus}$ with mutations in two key aromatic residues (W101A and F143A) had only slightly impaired binding to short ssDNA substrates (Supplementary Fig. 4b)[35,46]. Consistent with these findings, RPA-D$^{minus}$, which facilitated RPA to shift to the 20-nt mode, displayed similar on rate and final loaded amount as RPA-WT in our ssDNA Curtains experiments, but confers significant changes in the RPA ssDNA binding mode. Notably, Wold and colleagues demonstrated that RPA-D$^{minus}$ caused weak but reproducible defects in DNA repair, but not in DNA replication[46]. While the phenotype was puzzling due to comparable ssDNA affinity of RPA-D$^{minus}$ and RPA-WT, our findings indicate that the DNA repair phenotype of RPA-D$^{minus}$ was possibly caused by locking the bound RPA in the 20-nt ssDNA binding mode, thereby reducing ssDNA accessibility. We speculate that the cell may invoke an increased amount of the 20-nt mode of RPA ssDNA binding to ensure low ssDNA accessibility as a protection mode, while shifting RPA ssDNA binding from the 20-nt mode to the 30-nt mode triggers an action mode by increasing ssDNA accessibility to aid the loading of downstream DNA repair proteins like the Rad51 recombinase.

While the transition between the 20-nt and the 30-nt modes can be achieved by altering the salt concentration or RPA concentration in vitro, our study revealed the Rfa2-WH domain as one of the intrinsic regulatory elements of RPA ssDNA binding modes. The WH domain at the Rfa2 C-terminus, a major protein-interacting domain of RPA, is adjoined to DBD-D by a 42-aa linker region conferring its flexibility. Truncation of Rfa2 C-terminus leads to hypersensitivity of yeast cells to DNA damaging agents[47]. Although this phenotype could be attributed to its role in protein recruitment, our work suggests that the reduced ssDNA accessibility of RPA-ΔWH coated ssDNA can prevent the assembly of DNA repair factors, such as Rad51 (Fig. 6). Thus, the dynamic shift between RPA ssDNA binding modes may serve as a general regulatory mechanism for ssDNA accessibility. Previous biochemical experiments have demonstrated that RPA interaction capacity is essential for the mediator function of Rad52 in HR repair[43]. We further found that Rad52 is able to utilize its RPA-interacting middle acidic domain (Rad52-M) to orchestrate RPA ssDNA binding modes and RPA spacing, thereby regulating ssDNA accessibility and promoting Rad51 loading (Fig. 7). The Rad52-M domain is reported to interact

with both Rfa1 and Rfa2-WH[43]. Despite of the finding that Rfa2-WH plays an important role in RPA spacing regulation, our data suggested the presence of a second module outside of the Rfa2 WH domain which is subject to Rad52 regulation. It remains to be determined how Rad52-M operates through both interfaces. Besides Rad52, the cell hosts many other RIPs. It is interesting to investigate how these RIPs influence RPA dynamics to fulfill their molecular mechanisms. For instance, Rtt105 has been recently reported to interact with RPA like a chaperone and change RPA dynamic binding modes through induction of a longer extended ssDNA–RPA complex in vitro[19], and assures RPA assembly and high-fidelity DNA replication in vivo[48]. On basis of our RPA dynamic binding model, Rtt105 may facilitate RPA shifting to PBMs to adopt the protection mode with reduced ssDNA accessibility, a premise to be further tested in the future. Our in vitro data indicate that RPA ssDNA binding modes are governed by RPA concentration, providing another possible strategy for cells to control ssDNA accessibility by altering local RPA concentration. Consistent to this putative mechanism, forced exhaustion or elevation of cellular RPA level largely affect replication stress responses[49] and cellular RPA concentration increases approximately 2-fold during S phase[50]. These knowledges collected will not only elucidate the basic biology of RPA related cellular processes of DNA metabolic pathways, but may also enable therapeutic intervention in RPA related human diseases.

## Methods

### Construction of bacterial expression plasmids

The construct of GFP tagged *S. cerevisiae* RPA (scRPA) was kindly obtained from Dr. Hengyao Niu lab. *rfa1*, *rfa2*, and *rfa3* was cloned to the pET11c. To prevent the oligomerization of wild type GFP, we followed the previous references[51,52] to create a single mutation (A206K) on eGFP, which was named as MeGFP. MeGFP with a 6xHis tag was fused to the C-terminal of *rfa2* with a d(CTAGGC) linker (coding a Leucine and a Glycine). The construct of RPA-ΔWH-MeGFP was made by *rfa2*-Δ183-272 truncation. To express Rad52-M domain, truncation of scRad52 (205–295) was cloned to the pET21a with adding a His-tag and a 3◊Flag tag at N-terminus.

### Protein purification

We followed the previous references[9,28] to purify *S. cerevisiae* RPA. We transformed pET11c-RPA and pET11c-RPA *E. coli* BL21 codon plus cells. Four single colony was transferred to four flasks with 1 L LB at 37 degree in the shaker (without shaking). Because lucking of air, the OD$_{600}$ of the culture would not exceed 0.4 after overnight incubation. The next day, shake the culture until OD$_{600}$ reach 0.6. Around 1 mM IPTG was added to the culture and induced at 16 °C overnight. Pellets were collected at 5000 × *g*, re-suspended in a T-500 buffer (25 mM Tris-HCl (pH 7.5), 500 mM NaCl, 1 mM EDTA, 5% glycerol, 1 mM β-Mercaptoethanol (β-ME), 0.01% NP-40, and 1 mM PMSF) and sonicated for 10 min. Supernatant was collected after centrifugation at 20,000 × *g* for 30 min and then loaded to 10 mL Affi-Gel-Blue beads.

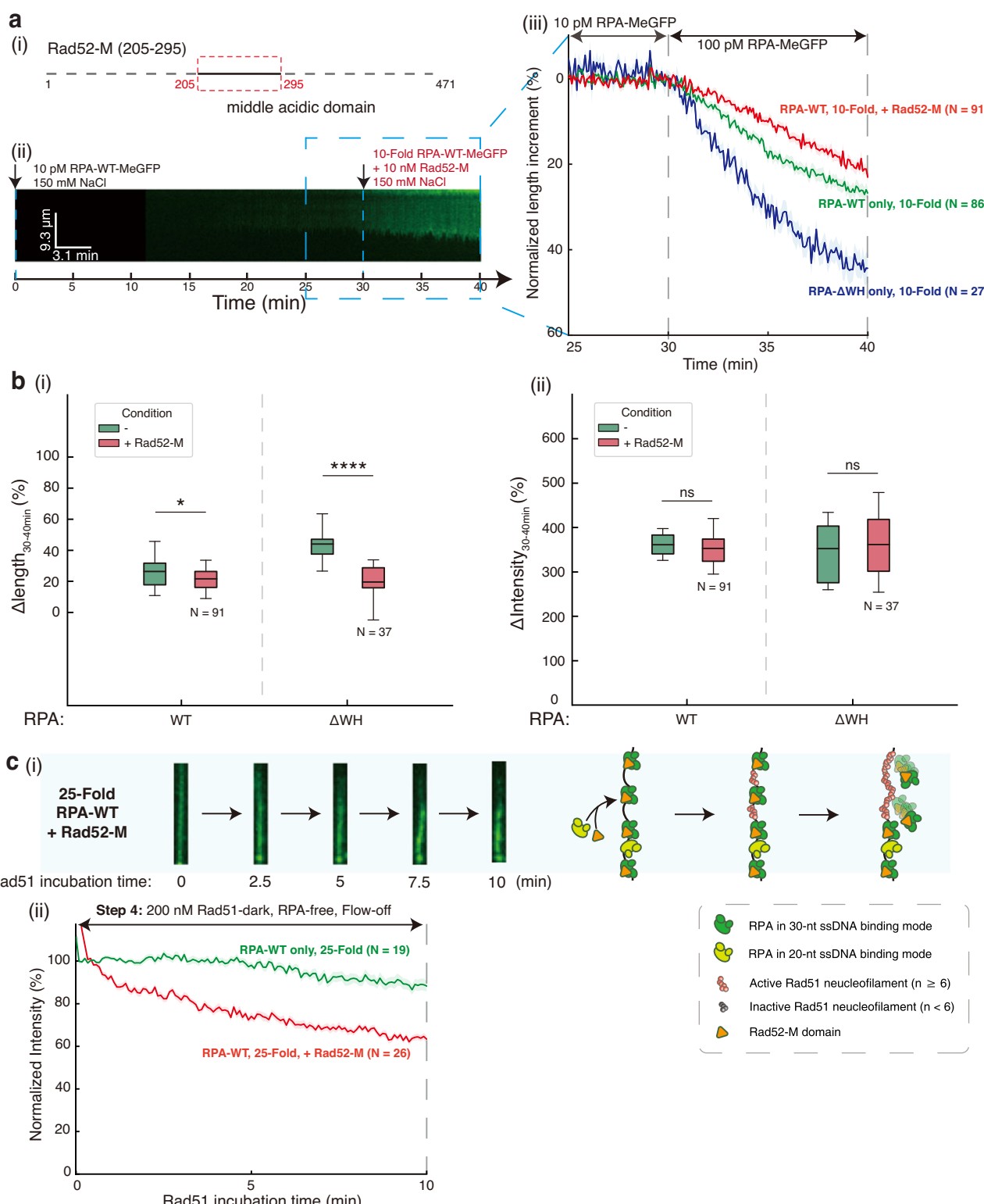

Washed the Affi-Gel-Blue beads with 50 mL T-500 buffer and 50 mL T-500 buffer plus 0.5 M NaSCN. Finally, the protein was eluted by an elution buffer (T-500 buffer plus 1.5 M NaSCN). The elution was applied to a Ni-NTA, and the beads were washed with 50 mL T-500 buffer and 50 mL T-500 buffer plus 20 mM imidazole. Finally, the protein was eluted by an elution buffer (T-500 buffer plus 200 mM imidazole). The elution was concentrated to 1.1 mL, and injected to Size Column (GE, Superdex® 200 10/300 GL) with a 1-mL loop in T-500 buffer. The fration can be tested by SDS-PAGE and Coomassie brilliant blue (CBB), trimer RPA will be eluted in 12 mL position. The pure trimer RPA will be collected and concentrated to 1 ml. Finally, we used the nanodrop to measure the absorption at 490 nm of RPA (Extinction Coefficien of GFP at 490 nm is 55,000 $M^{-1} cm^{-1}$). Purification of RPA-MeGFP and RPA mutants was the same. *S. cerevisiae* Rad51 were expressed and purified, as previously described[53]. For purification of the *S. cerevisiae* Rad52-M domain, pellets were collected at $5000 \times g$, re-suspended in a T-500 buffer (25 mM Tris-HCl (pH 7.5), 500 mM NaCl, 1 mM EDTA, 5% glycerol, 1 mM β-Mercaptoethanol (β-ME), 0.01%

**Fig. 7 | Interaction between Rad52 and RPA facilitates Rad51 nucleofilament formation by regulating RPA ssDNA binding mode and RPA spacing. a** Rad52-M domain affects the binding dynamics of RPA-WT. (i) schematic of Rad52-M domain, which was 205-345 of Rad52. Representative kymographs of 10-fold RPA/Rad52-M complex at 150 mM NaCl were shown in (ii), and its length analysis (blue) was in (iii), where length dynamic curve with 10-fold RPA-WT (green) from Fig. 3a and 10-fold RPA-ΔWH (red) from Fig. 6b at 150 mM NaCl were added for comparison. RPA/Rad52-M complex was prepared by 30-min pre-incubation. **b** Boxplot of length (i) or intensity (ii) change proportion within 30–40-min of ssDNA-RPA complexes from 10-fold RPA-WT, and RPA-ΔWH with (red) and without Rad52 M domain (green). Exact *p*-value (from left to right): 0.0128, 5. 3e-8, 0.2332, 0.7743. Boxplot style: middle line (mean), box range (0.25–0.75), whisker range (min-max), with outliers removed. Statistics: one-way ANOVA (analysis of variation). *P*-value style: GP: ≥0.05 (ns), <0.05 (\*), <0.01 (\*\*), <0.001 (\*\*\*), <0.0001 (\*\*\*\*). **c** RPA/Rad52-M complex facilitates Rad51 nucleofilament formation on ssDNA–RPA complex. Representative snapshots of ssDNA-RPA complex at 0, 2.5, 5, 7.5, 10 min after Rad51 injection were picked from double-tethered ssDNA Curtains results for 10-Fold RPA with Rad52 M domain (i). Quantification of GFP intensity decrease (ii) during 10-min Rad51 incubation for 10-Fold RPA with Rad52 M domain (red) in RPA-free working buffer and comparison with RPA-WT (green). Data are presented as mean ± SEM in **a**, **c**. *N* represents the total trace number of ssDNA-RPA complex end tracking examined over three times DNA Curtains experiments for each experimental condition in **a**–**c**. Source data are provided as a Source Data file.

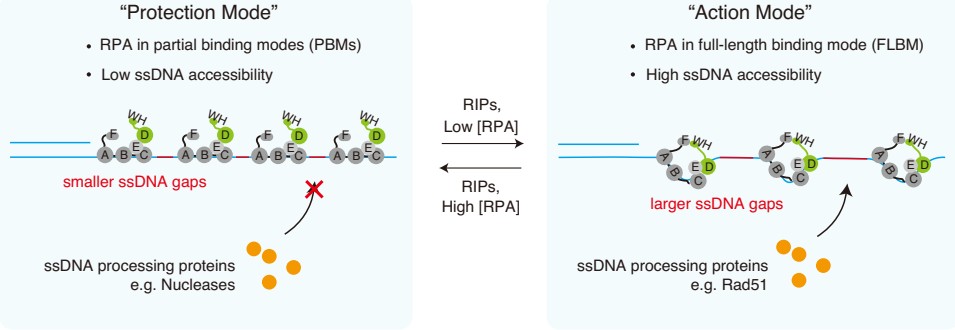

**Fig. 8 | A schematic concept of a biophysical mechanism involving multiple RPA binding dynamics on long ssDNA.** RPA ssDNA binding mode transition regulates RPA spacing on ssDNA, where RPA shifting from FLBM (30-nt mode) to PBMs (e.g. 20-nt mode) tightens the spacing between RPA molecules on ssDNA-RPA complex. Cell may evoke more PBMs of RPA binding to ensure low ssDNA accessibility as the protection mode in which RPA molecules tightly align on ssDNA to offer protection for the fragile ssDNA intermediates in replication forks, while that cell may inhibit RPA shifting to PBMs to provide high ssDNA accessibility as the action mode in which RPA binding leaves more uncovered ssDNA gaps to make room for loading of ssDNA processing proteins like the Rad51 recombinase during HR repair.

NP-40, and 1 mM PMSF) and sonicated for 10 minutes. Supernatant was collected after centrifugation at 20,000 × *g* for 30 min and then loaded to a Ni-NTA, and the beads were washed with 50 mL T-500 buffer and 50 mL T-500 buffer plus 20 mM imidazole. Finally, the protein was eluted by an elution buffer (T-500 buffer plus 200 mM imidazole), and stored at −80 °C in the T-500 buffer after further purification by gel filtration with a Superdex 200 column (GE Healthcare, USA).

### Electrophoretic mobility shift assay (EMSA)
For EMSA assays, ssDNA substrates with various length were radiolabeled on the 5′-end with [γ-32P] ATP using T4 Polynucleotide Kinase (New England Biolabs). In EMSA assays, RPA-WT and RPA mutants (RPA-dWH) were incubated with 5 nM 5′-radiolabeled 20-dT, 30-dT, 40-dT, 50-dT and 60-dT ssDNA substrates in 10 μL reaction buffer (20 mM Tris-HCl, pH 7.5, 150 mM KCl, 1 mM DTT, 100 μg/mL BSA). Alternatively, RPA-WT was also incubated with the same set of ssDNA substrates in 10 μL reaction buffer low-salt reaction buffer (20 mM Tris-HCl (pH 7.5), 15 mM KCl, 1 mM DTT, 100 μg/mL BSA) and the 15 mM KCl came directly from purified RPA and RPA mutants. The reactions were initiated by the addition of ssDNA substrates, kept at 30 °C for 10 min and terminated by moving the samples to room temperature. Samples were resolved on a 4% native polyacrylamide gel in 0.2× TBE buffer. The gels were kept in −20 °C freezer and developed overnight against phosphor-imager. On the following day, the phosphor-imager was scanned by Typhoon scanner (GE Healthcare).

### Microscale thermophoresis (MST) assays
MST assays were used to determine the binding affinity between RPA and dT30 ssDNA. Purified His-tagged RPA and His-tagged MeGFP-RPA were labelled with His-labeling dye according to the manufacturer's protocol (Monolith™ His-Tag Labeling Kit RED-tris-NTA 2nd Generation, NanoTemper Technologies, MO-L018). The dT30 ssDNA was diluted into a series of concentration gradient ranging from 0.003 nM to 100 nM in a 1×PBS-P+ buffer (diluted from 10×PBS-P+ buffer, cytiva, 280406), and mixed with labelled proteins at a volume ratio of 1: 1. All measurements were performed using a Monolith NT.115 device (NanoTemper Technologies) at medium MST power and 100% LED power, monitored by a MO.Control software(V1.6). Raw data were fitted with a Hill model to obtain the EC50 value.

### Three-step low-complexity ssDNA curtains
**Preparation of Low-complexity ssDNA substrates.** Base on the method developed before[27], we optimized the substrate design to avoid any wrong alignments and to raise the circular template formation efficiency. Template and primer oligos were purchased from RuiBiotech. For annealing, 4.5 μM biotinylated primer oligo (5′-/Biotin/-CAC CCA ACC ACC-3′) was annealed to 5 μM phosphorylated template oligo (5′-/Phos/-TGG GTG TGT GTG TGT GTG TGT GTG GTG GT-3′) in T4 DNA ligase buffer (B0202A, NEB) by first 5 min incubation at 90 °C and then slowly decreasing the temperature from 90 °C to the room temperature for 4 h. Then 0.5 μl 100 mM ATP and 1 μl T4 DNA ligase (M0202L, NEB) were added into the mix and the mix was incubated at room temperature for 4 h to ligate the nick in the annealed DNA circle. After the incubation, 450 μl EB buffer (QIAGEN) was added to dilute the annealed circles, which can be stored at 4 °C for ~1 month. Long ssDNA substrates for ssDNA Curtains was synthesized by rolling circle reaction in 1× RCR buffer (50 mM Tris-HCl (pH 7.5), 10 mM (NH$_4$)$_2$SO$_4$, 10 mM MgCl$_2$, and 4 mM DTT), 400 μM dNTPs (N0447L, NEB), 20 nM annealed circles, and 0.5 μM φ29 DNA polymerase (purified in-house). ssDNA synthesis was quenched by 10× dilution with working buffer (40 mM Tris-HCl (pH 7.5), 15 mM NaCl/150 mM NaCl, 2 mM MgCl$_2$, 1 mM DTT, and 0.2 mg/mL BSA).

**Total internal reflection fluorescence microscope (TIRFM).** All experimental data of DNA Curtains were acquired with a custom-built prism-type TIRFM (Nikon, Inverted Microscope Eclipse Ti-E), and the exposure time was 100-ms. The microscope was mounted with OBIS 488-nm LS 100-mW lasers. The real laser powers before the prism were measured: 488 nm, 3.2 mW (10%).

**Experimental setup.** The ssDNA Curtains was conducted as previously described[32,54]. The 10× diluted RCR product was injected into a flowcell with a flow rate of 0.03 ml/min. The working buffer was the BSA buffer, which was 40 mM Tris-HCl (pH 7.5), 15 mM NaCl/150 mM NaCl, 2 mM MgCl$_2$, 0.2 mg/mL BSA. For three-step design, single-tethered ssDNA substrates were first flushed with RPA-free working buffer, and switched to working buffer containing 10 pM RPA-MeGFP for 30 min flush with a flow rate of 0.4 ml/min at the second step, then at the third step, new working buffer containing RPA with different conditions was injected into the flowcell with a flow rate of 0.4 ml/min for another 10 min. At end of the experiments, flow was turned off to verify the mobility of ssDNA in the buffer flow.

### Image tracking analysis of DNA end

We track DNA end by a MeGFP signal tracking algorithm. We found the DNA end position directly by fitting the intensity array to a step function and optimized it by finding the point with maximal drop. We also tried several ways to determine the light and dark threshold, including using the average value or a clustering algorithm (The 'fminbnd' function in the Matlab software). After sorting the points by position, we calculated the variance of every three contiguous points, chose the three points with minimal variance, and took its average position as the final chosen position. Batch analysis on high-throughput data was realized by Python scripts.

### Statistics

All statistical significance values were evaluated based on one-way ANOVA (analysis of variation). $P$-value style: GP: $\geq 0.05$ (ns), $<0.05$ (*), $<0.01$ (**), $<0.001$ (***), $< 0.0001$ (****). Boxplot style: middle line (mean), box range (0.25–0.75), whisker range (min-max), with outliers removed.

### Reporting summary

Further information on research design is available in the Nature Portfolio Reporting Summary linked to this article.

## Data availability

The data that support the findings of this study are available from the corresponding author upon request. Source data are provided with this paper.

## Code availability

Image analysis was performed using Open source image processing software ImageJ (1.52 P) (http://imagej.net/Contributors), Python (3.7.0) (https://www.python.org), and MATLAB 2021a software (https://www.mathworks.com/products/matlab.html). The stochastic simulation model for multiple RPA molecules binding to long ssDNA was conducted by custom code in Julia (1.8.0) and is accessible can be accessed through https://github.com/hsianktin/RPA_model.

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

## Acknowledgements

We thank Dr. Letian Tao (Peking University, China) and Dr. Tom Chou (UCLA, USA) to discuss the Markov chain model in non-equilibrium physics. We thank Dr. Qing Li (Peking University, China), Dr. Xiaolan Zhao (Memorial Sloan-Kettering Cancer Center, USA), Dr. Shan Zha (Columbia University, USA), and the members of the Zhi Qi laboratory for comments on the manuscript. This work was supported by National Natural Science Foundation of China (Grant No. T2225009 (Z.Q.), 31670762 (Z.Q.), 32088101, and 22237002 (L.L.)), National Institute of Health (GM124765) and American Cancer Society Research Scholar Award (RSG-21-013-01-DMC) to H.N.

## Author contributions

H.N. and Z.Q. conceived, designed, and supervised the project. J.D. designed and performed biochemical experiments, single-molecule experiments, and data analysis, and wrote the manuscript. X.L. conducted theoretical simulation. J.S. performed biochemical experiments. Y.Z. assisted J.D. with single-molecule experiments. S.Z. and L.L. conducted the MST experiments. H.N. and Z.Q. analyzed the data and wrote the paper. All authors commented on the manuscript.

## Competing interests

The authors declare no competing interests.
