## [Peer Review File · Nature Communications]

ssDNA accessibility of Rad51 is regulated by orchestrating multiple RPA dynamicsREVIEWER COMMENTS

Reviewer #1 (Remarks to the Author):

The manuscript by Ding et al. examines the dynamics of multiple RPA binding to polymeric single stranded DNA using both single DNA molecule curtains and Markov chain modelling. They examine the effects of Rad52 mediator protein on ssDNA accessibility for Rad51 protein. They propose that Rad51 nucleation is regulated by transitions of RPA binding between two previously identified modes that occlude ~20 ("protection mode") vs. ~30 nucleotides ("action" mode) on ssDNA. They further suggest that the Rfa2 WH domain facilitates the protection mode while Rad52 inhibits this mode.

In general, it is commendable that the authors attempt to understand this complex system quantitatively. The effects of a transition between the two different RPA binding modes may be important functionally. However, the system is very complex, even just for RPA binding when the two modes are considered. My enthusiasm for the ms. is diminished based on three considerations.

1. My main concern is the use of GFP-tagged RPA. The authors do not present a careful comparison of the binding properties of the GFP-RPA compared to wild type un-labeled RPA. They only show a qualitative comparison (Supplementary Figure 2b vs. Figure 1bii) indicating that RPA-GFP can bind dT30 by an EMSA assay. They need to compare the binding quantitatively. They need to show a full isotherm of the tagged RPA compared to untagged RPA using a more rigorous assay than an EMSA assay. They also need to examine whether the binding mode transitions of RPA are influenced by the GFP tag. I also was unable to determine whether the GFP was at the N- or C-terminus of RPA.

2. The general conclusions based on the curtains data make qualitative sense in terms of the likely effects of an RPA binding mode transition. However, I am skeptical that there is sufficient information available to quantitatively describe the curtains data using the model that the authors describe. This is partly because the stochastic modelling and the curve fitting are not explained in sufficient detail.

The model used to describe the curtains data contains 6 parameters without considering cooperativity (Supp Fig. 5e). It is not clear to me how well these parameters can be determined. The authors provide some statistical analysis in Supp Fig 5e, but do not provide error estimates of the individual parameters. How well are these constrained?

It is not clear how or if overlap of sites is included in the model? Various papers have claimed that RPA does and does not bind cooperatively. The authors do not seem to include cooperativity. Does the binding mode affect cooperativity?

The fitted simulation parameters used to describe the curves (shown in Supp Figure 5e) need to be interpreted. What are the actual bimolecular rate constants, k_1 , in conventional units of $M^{-1} s^{-1}$? Do they make sense? How do these parameters compare with experimental estimates of the rate constants and equilibrium binding constants?

RPA is capable of diffusing along ssDNA. However, this kinetic feature does not appear to be considered in the stochastic model?

3. The authors do not appear to have enough information about Rad51 or Rad52 binding parameters to perform the simulations that they show for RPA in the presence of these additional proteins.

Minor comments

The three step low-complexity ssDNA Curtains method described here does not appear to be novel.

Page 7, line 5: The term ionic strength is not appropriate in discussions of protein-nucleic acid interactions in general. The RPA-ssDNA interaction is influenced by the salt concentration and type (valence) and is not a function of the ionic strength.

Page 7, lines 3-5: I am confused the author's implication that non-saturating conditions are non-equilibrium conditions, whereas near saturating conditions are equilibrium conditions. This makes little sense. The system can be at equilibrium under any level of saturation.

Reviewer #2 (Remarks to the Author):

In this manuscript, Ding et al describe the dynamics of RPA in binding ssDNA, and regulatory mechanisms exerted by RPA domains and the recombination mediator Rad52 with a combination of biochemical assays, single-molecule analysis (TIRFM), and molecular simulation. First, they show that RPA adopts alternate DNA binding modes in a DNA length/RPA concentration-dependent manner. From DNA curtain analysis, the authors observe salt-dependent changes in the size of RPA-DNA complexes and suggest that the size difference reflects adoption of the 20-nt or 30-nt binding mode by RPA. These results are further supported by molecular simulation data predicting the length of ssDNA gap between bound RPA molecules to allow for nucleation of the recombinase Rad51 and assembly of the Rad51-ssDNA nucleoprotein filament capable of DNA homology search and strand exchange. The authors also show that OB fold-D in Rfa2 helps mediate the conversion from a "protective" to "action" DNA binding mode, with the latter mode being more compact with larger DNA gaps in-between nucleoprotein complexes and thus conducive for nucleating Rad51 nucleoprotein filament formation. Then, data are presented to show opposing roles of the Rfa2-WH (winged helix) domain and Rad52-M RPA binding domain in regulating the compactness of RPA-ssDNA complexes, ssDNA gaps in nucleoprotein complexes, and Rad51 nucleation.

RPA interacting proteins (RIPs), such as yeast Rad52, are known to regulate DNA access by client proteins, Rad51 in this case. However, molecular details of how RPA-Rad51 handoff occurs are lacking still. This study combining biochemical, single-molecular biophysical, and simulation analyses goes a significant distance in providing these details.

1. Please quantify the key EMSA data. Moreover, authors' claim that a second RPA protomer can bind the dT-52 substrate at a lower RPA concentration than dT-40 is not immediately apparent from the data shown.
2. EMSA should be carried out at 15 mM NaCl with a few key DNA substrates. In addition, DNA binding by the two RPA variants (D-minus and delta WH) need to be examined by EMSA.
3. Mass photometry, if available, would be quite useful for distinguishing between nucleoprotein complexes harboring one or more RPA protomers.
4. The blue dots in Fig. 2e(i) are difficult to see.
5. The results in Fig 3c & Fig 3S-d appear to show that RPA amounts loaded on ssDNA are comparable at both salt concentrations. However, the simulation results seem to indicate that more RPA molecules associate with DNA at the lower salt concentration. Please reconcile these results.

6. Fig 6c: results of Δ -WH at 15 mM NaCl should be included if available.

7. There are grammatical errors throughout the manuscript. Please proof the revised manuscript carefully before resubmission.

Reviewer #3 (Remarks to the Author):

RPA is a heterotrimeric ssDNA binding protein that plays important roles in all aspects of DNA metabolism involving an ssDNA intermediate, including the repair of DNA damage through homologous recombination (HR). An interesting problem with RPA is that in cells, RPA is one of the first proteins to bind to early ssDNA intermediates during HR and it binds to ssDNA much more tightly than the recombinase Rad51, which is the key protein necessary for catalyzing the DNA strand exchange reactions that take place in HR. Full mechanistic details of how a relatively weak ssDNA binding protein (Rad51) manages to replace a tight ssDNA binding protein (RPA) are lacking within the field. In cells this process requires the involvement of mediator proteins such as Rad52 (in yeast) or BRCA2 (in humans), but how they accomplish this replacement reaction is not really understood other than at a very basic level.

In this study, Ding et al. use a combination of bulk biochemical, single molecule and computational simulations to investigate the properties of RPA-ssDNA complexes to try to establish how their biophysical properties might facilitate the early stages of Rad51 filament formation. The result of this work is a new quantitative model that describes how the dynamic transitions between distinct RPA-ssDNA-binding modes can themselves contribute to Rad51 filament formation, and how the mediator protein Rad52 can assist in this process. The authors propose that RPA exists in a "protection mode" that enables it to protect ssDNA from potentially damaging nucleases and a "action mode" allowing for its replacement by Rad51.

The experiments are well designed, the data are technically good, the results in general support the authors model (with some caveats) and will also generate interest within the field. The manuscript does however require extensive English language editing before it can be published. My scientific comments (below) are relatively minor and focused more on some clarifications.

Comments:

Figure 2: Most readers will not know the significance of panel iii in figure 2d. This should be explained in the main text or in the figure legend. In figure 2e panel ii, the PI will need to clarify that the trace represents just one ssDNA molecule, and it would be beneficial if the error bars were shown (in the legend, it states that "Error bars, mean \pm s.e.m., but I don't see any error bars). Be sure to explain that in subsequent figures the traces represent individual ssDNA molecules, and the boxplots represent the collective data.

Figure 3: The side-by-side presentation of the data sets in Figure 3c and 3d seem a bit confusing. Is there some way to emphasize that both bars represent the same Fold RPA? This also applies to other figures with boxplots.

Regarding the simulations, the y-axes are labeled as "normalized length increments", which I presume to be derived from the value of $xr(t)$, and the resulting simulation data is intended to match the presentation of the single molecule experiments. It is not clear to me how the simulation length increments account for each of the three possible states of the bound ssDNA (naked ssDNA, RPA in the 20-nt mode and RPA in the 30-nt mode). My impression is that each of these states must have an assigned length value within the simulation and that they must also account for the fact that they are being compared to ssDNA molecules held in an extended configuration within a lamina flow system. If my interpretation is correct, what are the mean extended length values for each aforementioned

state? I suppose this corresponds to the “m” values in Figure 4a, correct? If so, what are the numerical values for “m”, and do they make physical sense with respect to ssDNA characteristics in the flow stretched system?

An important potential issue with the authors’ model is that it assumes that there is an absolute requirement for an ≥ 18 -nt ssDNA gap to allow for the initial nucleation of a Rad51 filament and it does not consider the possibility that shorter ssDNA gaps may allow for nucleation events wherein further addition of nucleating monomers promotes the release or partial release of RPA ssDNA binding domains. For example, a 15 nt gap would allow for 5 Rad51 monomers, and a 6th could easily bind if one of the RPA OB-folds transiently dissociated from the ssDNA, or if RPA were to diffuse a short distance along the ssDNA (transient domain dissociation and 1D diffusion are both known characteristics of RPA). Similar arguments could be made for even smaller gaps. These features are not accounted for in the computer simulations or in the authors’ model. I do not think that these issues invalidate the authors’ general model (i.e., transitions between distinct RPA binding modes may affect the fraction of ssDNA that is accessible for binding by other proteins), nor do I think they need to be experimentally or computationally addressed at this stage (probably too difficult), but they should at least be clear acknowledged in the Discussion.

POINT BY POINT RESPONSE TO REVIEWER COMMENTS

Reviewer #1 (Remarks to the Author):

The manuscript by Ding et al. examines the dynamics of multiple RPA binding to polymeric single stranded DNA using both single DNA molecule curtains and Markov chain modelling. They examine the effects of Rad52 mediator protein on ssDNA accessibility for Rad51 protein. They propose that Rad51 nucleation is regulated by transitions of RPA binding between two previously identified modes that occlude ~20 (“protection mode”) vs. ~30 nucleotides (“action” mode) on ssDNA. They further suggest that the Rfa2 WH domain facilitates the protection mode while Rad52 inhibits this mode.

In general, it is commendable that the authors attempt to understand this complex system quantitatively. The effects of a transition between the two different RPA binding modes may be important functionally. However, the system is very complex, even just for RPA binding when the two modes are considered. My enthusiasm for the ms. is diminished based on three considerations.

We thank the reviewer for these important comments and questions, and we are glad to hear that the reviewer appreciates the complexity of the system we chose to study. New control experiments and simulations have been conducted, and the relevant text and references have been revised according to the reviewer’s suggestions (see below). We also added some of these thoughts inspired by the reviewer into the discussion part in the revised manuscript. All changes in the manuscript text file were highlighted in red.

1. My main concern is the use of GFP-tagged RPA. The authors do not present a careful comparison of the binding properties of the GFP-RPA compared to wild type un-labeled RPA. They only show a qualitative comparison (Supplementary Figure 2b vs. Figure 1bii) indicating that RPA-GFP can bind dT30 by an EMSA assay. They need to compare the binding quantitatively. They need to show a full isotherm of the tagged RPA compared to

untagged RPA using a more rigorous assay than an EMSA assay. They also need to examine whether the binding mode transitions of RPA are influenced by the GFP tag. I also was unable to determine whether the GFP was at the N- or C-terminus of RPA.

We thank the reviewer for these important comments and suggestions. We have conducted new control experiments of MicroScale Thermophoresis (MST) assays for the full isotherm of RPA-MeGFP compared to RPA (Reviewer Only Fig. 1) and have now included the data into Supplementary Fig. 2c-d. Quantification of the MST results demonstrated that RPA and RPA-MeGFP bound to dT30 with comparable binding properties. In our experiments, the GFP tag was added at C-terminus of Rfa2, which has been studied before in DNA curtains analysis proved to have minimal impact on its activities¹⁻³. Previous ssDNA Curtains experiments have demonstrated that the exchange between RPA and RPA-GFP on long ssDNA substrates have no impact on the length of ssDNA-RPA complexes, which suggested the GFP tag had minimal impact on the RPA binding mode transitions⁴. We fully understand the concerns in using GFP tag to proteins and we have revised the relevant text to better elucidate the properties of RPA-MeGFP we used.

Reviewer Only Figure 1 (included into Supplementary Fig. 2c-d). MST assays suggested RPA and RPA-MeGFP bound to dT30 with comparable binding properties. (a) RPA-MeGFP. (b) RPA.

2. The general conclusions based on the curtains data make qualitative sense in terms of

the likely effects of an RPA binding mode transition. However, I am skeptical that there is sufficient information available to quantitatively describe the curtains data using the model that the authors describe. This is partly because the stochastic modelling and the curve fitting are not explained in sufficient detail.

The model used to describe the curtains data contains 6 parameters without considering cooperativity (Supplementary Fig. 5e). It is not clear to me how well these parameters can be determined. The authors provide some statistical analysis in Supplementary Fig. 5e, but do not provide error estimates of the individual parameters. How well are these constrained?

It is not clear how or if overlap of sites is included in the model? Various papers have claimed that RPA does and does not bind cooperatively. The authors do not seem to include cooperativity. Does the binding mode affect cooperativity?

The fitted simulation parameters used to describe the curves (shown in Supplementary Fig. 5e) need to be interpreted. What are the actual bimolecular rate constants, k_1 , in conventional units of $M^{-1} s^{-1}$? Do they make sense? How do these parameters compare with experimental estimates of the rate constants and equilibrium binding constants?

RPA is capable of diffusing along ssDNA. However, this kinetic feature does not appear to be considered in the stochastic model?

We thank the reviewer for the comments, which are all great points for discussion. We are sorry for the confusion partially caused by the lack of sufficient details in the Main Text section on the modeling process, parameter interpretation and parameter determination process (originally placed in Supplementary Methods), and the error

estimation of parameters (originally shown in Supplementary Fig. 5), due to space limitation. We have now revised the relevant text accordingly to the reviewer's comments, which help us to better clarify the simulation model and to discuss the issues of cooperativity and diffusion. We also endowed RPA with diffusion property in our new model and conducted new simulations to answer the question of diffusion. Our response to the reviewer's inquiries on 1) Quantitative modeling principle; 2) Parameter determination and error estimation; 3) Physical sense of the kinetic parameters; 4) Site overlapping; 5) Cooperativity and 6) Diffusion are depicted below.

Quantitative modeling principle

Our model aimed to provide a quantitative description of the sequential binding and mode transition of RPA on linearized ssDNA interface with a finite length, based on our biophysical understanding of the underlying molecular processes. Specifically, we modeled the following processes and interactions:

- Nonspecific binding between ssDNA and RPA.
- Volume exclusion. Any nucleotide occupied by RPA will not be accessible to other RPA or proteins for binding.
- RPA mode switching after binding, toggling between 20-nt mode and 30-nt mode. The initial configuration is always taken to be 20-nt mode. Different modes have different DNA footprints. The processes of RPA binding and RPA mode switching are depicted by four kinetics parameters (k_1 , k_{-1} , k_2 , k_{-2}).
- Length extension of RPA-ssDNA complex. RPA in different modes are considered to reside in different conformation and thus can extend DNA to different length. This property is depicted by the two length conversion parameters α and β .

- We added diffusion to both modes in the form of symmetric one-dimensional random walks on the ssDNA. For the sake of simplicity, we have assumed that both modes have the same.
- RPA cooperation is not considered in our model.

Parameter determination and error estimation

For the parameter determination, we adopted stochastic gradient descent method to minimize the weighted mean squared loss between the length dynamics curves from curtain experiments and from stochastic simulations.

We cannot directly provide the confidence intervals for the parameters. This is because our parameter determination method is not likelihood-based. As a result, there is no statistically well-defined way to compute the confidence interval for each parameter. Given the high computational burden for each individual simulation and the combinatorically large state space, likelihood-based method like Expectation-Maximization (EM) method is infeasible for our model.

To provide an alternative to the confidential level, we presented Supplementary Fig. 5 for error estimation and also to explain the sensitivity of our model to the parameters (Reviewer Only Fig. 2, which is cropped from panel a-f of Supplementary Fig. 5). Among the six aforementioned parameters, the binding rate for 20-nt mode, k_1 , is the most sensitive parameter and well confined. The rest of the parameters interact with each other, making some part of the error curves flat, meaning the inability of us to further confine the region of parameters. Insensitivity to certain parameters is termed sloppiness and is widely observed for many complex biological systems on various levels⁵. This insensitivity allowed us to capture the dynamics without tremendous efforts to find a set of truly optimal parameters.

Reviewer Only Figure 2 (cropped from panel a-f of Supplementary Fig. 5). Error estimation of the six simulation parameters.

Physical sense of the kinetic parameters

To compare the parameters with existing measurements, we converted the kinetic rates resulted from our model into apparent dissociation constants. The kinetic parameters estimated by the simulations (Reviewer Only Table 1) were comparable to

the reported Kd of scRPA binding to ssDNA (6.3×10^{-11} M under 20 mM NaCl condition and 1.1×10^{-9} M under 200 mM NaCl condition)⁶.

Reviewer Only Table 1: Kinetic parameters estimated by the simulations

	Kd_{20-nt mode}	Kd_{30-nt mode}
wt 15mM NaCl	1.8×10^{-11} M	1.0×10^{-13} M
wt 150mM NaCl	0.6×10^{-10} M	1.3×10^{-12} M

The kinetic parameters of RPA in 20-nt mode (for short in the formula, 20nt) were estimated by the following considerations:

$$k_{on}(20nt) \times c_0 = k_1, \quad k_{off}(20nt) = k_2$$

$$k_{on}(20nt) = k_1/c_0, \quad k_d(20nt) = \frac{k_{off}(20nt)}{k_{on}(20nt)}$$

For the binding affinity of the full-length 30-nt mode (for short in the formula, 30nt), we have the following estimates:

Since 20-nt to 30-nt transition would be in quasi-steady state. Under quasi-steady state, the probability that RPA is in 20-nt mode is given by:

$$P(20nt) = \frac{k_{open}}{k_{open} + k_{close}} = 1 - P(30nt)$$

We treat the complex as a whole, and the system reduces to simple binding kinetics.

$$\tilde{k}_{on} = k_{on}(20nt), \quad \tilde{k}_{off} = k_2 \cdot P(20nt)$$

This allows us to define the dissociation constant by:

$$\tilde{K}_d = \tilde{k}_{on}/\tilde{k}_{off} = K_d(20nt) \cdot P(20nt)$$

since the complex leaves the system only if it is in 20-nt mode.

In our model, the precision estimates of $k_2 = k_{off}$ are limited by the time-span of ~ 1000 s. The length extension-time curve would look similar for different small values $\tilde{k}_{off} = k_2 \cdot P(20nt)$ as long as it is something much smaller than 10^{-5} s^{-1} .

As is demonstrated experimentally, we cannot observe significant decrease in ssDNA-RPA length or RPA intensity kept on ssDNA after switching to blank buffer. This suggests a very small \tilde{k}_{off} and limits our ability to constrain k_2 nicely. Above arguments also explain the flat loss curves observed in k_2 .

Site overlapping

Site overlapping of RPA is prohibited in our model due to volume exclusion. In our model, when the 30-nt mode RPA switches to 20-nt mode, a 10nt ssDNA will be released for binding with other proteins (Reviewer Only Fig. 3, which is cropped from panel b of Fig. 4). Since we consider the 30-nt mode as the full-length RPA binding mode, then this situation could be counted as “partial overlapping” in a broad sense. Nevertheless, any nucleotide would never be occupied at the same time in our model.

Reviewer Only Figure 3 (cropped from panel b of Fig. 4). Illustration for RPA mode transition and ssDNA gap.

Cooperativity

We thank the reviewer again for this important topic and please find our response below. In addition, we have revised the discussion section to incorporate some of the thoughts inspired by this question

There were plot twists and turns in the study of RPA cooperativity. In the 1990s, human RPA (hRPA) was shown to have an occluded binding site size of ~30 nt and bound ssDNA with low cooperativity ($\omega = \sim 10-20$)⁷. RPA homologue from *D. melanogaster* shows similar low cooperativity binding pattern with hRPA⁸. RPA from *S. cerevisiae* (scRPA) was initially shown to have an occluded binding site of 90–100 nt and to bind with very high cooperativity. Later, many independent researchers reported the occluded binding site of scRPA to be between 20 and 30 nt^{6,9,10} and the cooperativity of scRPA to be low^{6,11}. Our fine size titration using dT10-60 by EMSA also supports the high affinity and low cooperativity binding pattern of scRPA (Fig. 1). For example, our data showed that when binding to 40 nts ssDNA substrate, the stable binding of first RPA and second RPA occur when RPA concentration reached 5-10 nM and 50-100 nM, respectively, indicating there is no significant positive cooperativity between the two RPA binding events. The observations of ssDNA gaps on RPA-ssDNA complex in DNA Curtains analysis also supports the low cooperative binding pattern of RPA in solution environment. We thus followed the mainstream view and made the elementary hypothesis that RPA has no cooperativity in our simulation model in current stage.

There is no denying that the potential cooperative binding under certain conditions is an important property of RPA and an interesting research direction. Recently, CryoEM and FRET studies have reported that S178 phosphorylation regulates scRPA binding to ssDNA by inducing cooperativity¹². We believe the cells can adopt posttranslational modifications and possibly some other strategies to induce RPA cooperative binding to fulfill its function. In future, we would like to take the advantage of the platform that we devised to study how RPA cooperativity contributes to its binding dynamics and biological function.

Diffusion

We thank the reviewer again for this important topic. To answer this question, we have added diffusion to our model and conducted new simulations for the 1D diffusion of RPA along the DNA. We found that adding diffusion to our model did not significantly alter the binding dynamics of RPA.

In short, we model it as a symmetric 1D random walk with simple exclusion effects. We noted that the diffusion coefficient for hRPA has been measured to be $> 10^3$ nt²/s and we did not find any literatures reporting diffusion coefficient for scRPA¹³. However, fast diffusion makes the exact Gillespie simulation intractable.

Fortunately, adding diffusion to our model does not significantly affect the dynamics of the RPA-DNA interaction, as shown in Reviewer Only Fig. 4. We first conduct a simulation with varying diffusion coefficients from 10^{-6} nt²/s to 10^0 nt²/s, while keeping other parameters the same (Reviewer Only Fig. 4a). We have found that the L2 norm of the simulated traces is insensitive to the diffusion coefficient. Furthermore, we reduced the total length of DNA to 100 nt and experimented with larger diffusion coefficients up to 10^2 nt²/s (Reviewer Only Fig. 4b). The same level of insensitivity was observed again.

Reviewer Only Figure 4. The difference from experimental value (loss function, L2 norm) of the simulated traces is insensitive to the diffusion coefficient. (a) The diffusion coefficient is varied from 10^{-6} nt²/s to 10^0 nt²/s, while keeping other parameters the same as the ones obtained by the gradient descent method. **(b)** We further increase the diffusion coefficient to 10^2 nt²/s and reduce the total length of DNA to 500bp. The L2 norm of the simulated traces is still insensitive to the diffusion coefficient.

Given that the diffusion around 10^2 nt²/s is much faster than the binding and conformation switching rates in that system, we therefore conclude that even in the fast diffusion regime, the dynamics of the RPA-DNA interaction is similar to the slow diffusion regime.

This conclusion does not negate the role of diffusion and can possibly be interpreted as following. The primary role of diffusion is to change the position of a DNA-bound RPA. However, unbinding and rebinding of an RPA is equivalent to diffusion in terms of moving RPAs around. Consequently, introducing diffusion does not introduce new physics into our model. Since the diffusion coefficient of yeast RPA remains unknown, we chose not to include this data in the revised manuscript and will revisit this in future studies. Hopefully these new results could eliminate the concerns on the influence of diffusion to RPA binding properties.

3. The authors do not appear to have enough information about Rad51 or Rad52 binding parameters to perform the simulations that they show for RPA in the presence of these additional proteins.

We thank the reviewer for these comments and questions. We are sorry that we did not perform simulations for the binding of RPA in the presence of Rad51 or Rad52, due to limited knowledge of RPA interaction dynamics with Rad51 or Rad52 in this field.

These sets of Monte Carlo simulations were performed to fit the binding curve from DNA Curtains analysis for RPA-WT only. Our model aimed to simulate the bimolecular dynamic behaviors of RPA on long ssDNA, which broadens our knowledge in this

complicated system of multiple RPA binding dynamics. It would be impossible to set up such a model without numerous literatures for the basic biochemical and biophysical properties of RPA, e.g. 20-nt mode and 30-nt mode. In future, we wish we could incorporate other RPA-interacting proteins like Rad52 into our RPA dynamic binding model. However, in current stage, there is no first principle for RPA-Rad52 interaction. We are happy to see several works trying to decipher the biophysical nature of RPA-Rad52 interaction ¹⁴. In the future, it will be very interesting to examine how interaction with Rad52 change the conformational dynamics of RPA-ssDNA complex, and set up a model of RPA dynamic binding in presence of Rad52 when more structural and biophysical studies on this interface become available.

Because there is no direct interaction between Rad51 and RPA reported, we did not obtain any RPA binding curves in the presence of Rad51 and thus cannot perform corresponding simulations. Eric C. Greene and his coworkers studied the nucleofilament assembly and disassembly process of hRad51 alone or in the presence of hRPA ¹⁵. However, it is unclear that how RPA binding dynamics is affected by the competition of Rad51. It would be very interesting to study RPA binding dynamics on long ssDNA in presence of Rad51 in the future.

Minor comments

The three step low-complexity ssDNA Curtains method described here does not appear to be novel.

We thank the reviewer for the comments. We revised the relevant text to avoid using such words like novel. The ssDNA Curtains method was initially developed by Eric C. Greene group ¹⁶, and Ilya J. Finkelstein group applied low-complexity substrates to ssDNA Curtains ⁴. We applied the three-step experimental design to the low-complexity ssDNA Curtains combined with length analyses, which provided a synchronized initial state of ssDNA-RPA complex to monitor RPA dynamics on long ssDNA substrates starting from a non-equilibrium state.

Page 7, line 5: The term ionic strength is not appropriate in discussions of protein-nucleic acid interactions in general. The RPA-ssDNA interaction is influenced by the salt concentration and type (valence) and is not a function of the ionic strength.

We thank the reviewer for these comments and questions. We fully agree that using salt concentration is more appropriate and precise in discussions of RPA-ssDNA interaction and will lead to less confusion. We revised the relevant text to use “salt concentration” instead of “ionic strength”.

Page 7, lines 3-5: I am confused the author’s implication that non-saturating conditions are non-equilibrium conditions, whereas near saturating conditions are equilibrium conditions. This makes little sense. The system can be at equilibrium under any level of saturation.

We thank the reviewer for these comments and questions. We agree that the system can be at equilibrium under any level of saturation. We are sorry for the confusion and we have now revised the relevant text to avoid raising confusion in these concepts.

Reviewer #2 (Remarks to the Author):

In this manuscript, Ding et al describe the dynamics of RPA in binding ssDNA, and regulatory mechanisms exerted by RPA domains and the recombination mediator Rad52 with a combination of biochemical assays, single-molecule analysis (TIRFM), and molecular simulation. First, they show that RPA adopts alternate DNA binding modes in a DNA length/RPA concentration-dependent manner. From DNA curtain analysis, the authors observe salt-dependent changes in the size of RPA-DNA complexes and suggest that the size difference reflects adoption of the 20-nt or 30-nt binding mode by RPA. These results are further supported by molecular simulation data predicting the length of ssDNA gap between bound RPA molecules to allow for nucleation of the recombinase Rad51 and assembly of the Rad51-ssDNA nucleoprotein filament capable

of DNA homology search and strand exchange. The authors also show that OB fold-D in Rfa2 helps mediate the conversion from a “protective” to “action” DNA binding mode, with the latter mode being more compact with larger DNA gaps in-between nucleoprotein complexes and thus conducive for nucleating Rad51 nucleoprotein filament formation. Then, data are presented to show opposing roles of the Rfa2-WH (winged helix) domain and Rad52-M RPA binding domain in regulating the compactness of RPA-ssDNA complexes, ssDNA gaps in nucleoprotein complexes, and Rad51 nucleation.

RPA interacting proteins (RIPs), such as yeast Rad52, are known to regulate DNA access by client proteins, Rad51 in this case. However, molecular details of how RPA-Rad51 handoff occurs are lacking still. This study combining biochemical, single-molecular biophysical, and simulation analyses goes a significant distance in providing these details.

We thank the reviewer for these important comments and questions, and we are glad to hear that the reviewer appreciates the complexity of the system we chose to study. New control experiments and simulations have been conducted, and the relevant text and references have been revised according to the reviewer’s suggestions (see below). We also added some of these thoughts inspired by the reviewer into the discussion part in the revised manuscript. All changes in the manuscript text file were highlighted in red.

1. Please quantify the key EMSA data. Moreover, authors’ claim that a second RPA protomer can bind the dT-52 substrate at a lower RPA concentration than dT-40 is not immediately apparent from the data shown.

We thank the reviewer for the important comments and suggestions. We have quantified the key EMSA data (Reviewer Only Fig. 5) and have now included the results into Fig. 1d. While the dT50 remains to be more prone to the second RPA protomer binding, there is a big error range. We have therefore removed this claim. Overall, the revised EMSA data support our conclusion that 20-nt mode and the 30-nt mode are relatively stable and elevated RPA concentration is able to induce mode change from 30-nt to 20-nt mode.

Reviewer Only Figure 5 (included into Fig. 1d). Quantification of key EMSA data. (a) Quantification of RPA binding percent on the titration of RPA to key ssDNA substrates. **(b)** Quantification of 2nd RPA binding percent on the titration of RPA to key ssDNA substrates.

2. EMSA should be carried out at 15 mM NaCl with a few key DNA substrates. In addition, DNA binding by the two RPA variants (D-minus and delta WH) need to be examined by EMSA.

We thank the reviewer for the important suggestions. We have conducted new EMSA experiments suggested by the reviewer (Reviewer Only Fig. 6) and have now included the results into Supplementary Fig. 1b, Supplementary Fig. 4b, and Supplementary Fig. 7a. The DNA binding by RPA at 15 mM NaCl, RPA-D^{minus} and RPA-ΔWH at 150 mM NaCl, together with by RPA at 150 mM NaCl are comparable in EMSA assays.

Consistent with previous studies¹⁷, the resulting RPA-D^{minus} did not significantly altered the binding of RPA to short ssDNA substrates in EMSA, which could be due to the high affinity of RPA in ssDNA binding and the limitation of sensitivities in bulk assays. Similar reasons can account for the behaviors of RPA-ΔWH.

Reviewer Only Figure 6 (included into Supplementary Fig. 1b, Supplementary Fig. 4b, and Supplementary Fig. 7a). EMSAs of RPA binding to poly-dT ssDNA substrates with various length. (a) RPA at 15 mM NaCl. (b) RPA-D^{minus} at 150 mM NaCl. (c) RPA- Δ WH at 150 mM NaCl.

3. Mass photometry, if available, would be quite useful for distinguishing between nucleoprotein complexes harboring one or more RPA protomers.

We thank the reviewer for this suggestion. Mass photometry can measure the mass of macromolecule without labeling under nondegenerative conditions and we agree that it can be a powerful tool for distinguishing protein-DNA complex in different states. We investigated searchable information of the mass photometry, unfortunately, we found it unavailable among our accessible resources. We will conduct the suggested analysis once we could get access to this instrument in the future.

4. The blue dots in Fig. 2e(i) are difficult to see.

We thank the reviewer for this comment and suggestion. We are sorry that the blue dots in Fig. 2e(i) faded in color due to some issues during image export. We have now revised the Fig. 2 to make sure that the blue dots can be clearly seen.

5. The results in Fig 3c & Fig 3S-d appear to show that RPA amounts loaded on ssDNA are comparable at both salt concentrations. However, the simulation results seem to indicate that more RPA molecules associate with DNA at the lower salt concentration. Please reconcile these results.

We thank the reviewer for these important comments and suggestions. We are sorry that the data presentation of simulation results could lead to confusion when comparing to the results of intensity analyses, because it was the normalized intensity increment

during 30-40 min that be plotted in Fig. 3c, while it was the absolute number of loaded RPA at 40-min that be listed in Supplementary Fig. 6b. We have conducted new analyses on the existing simulation data, and the simulated loaded RPA numbers were comparable to the intensity analysis from experimental results (Reviewer Only Fig. 7). We have now included the results into Supplementary Fig. 6b.

We note that the intensity analysis of experimental results in Fig. 3c and Supplementary Fig. 3d used normalized intensity value, which presented the relative intensity increment based on the intensity levels at 30 min. We have plotted the number of newly loaded RPA during 30-40 min from the simulation results (Reviewer Only Fig. 7). We then apply independent t-test to compare the difference of RPA number increment from 30-min to 40-min ($\Delta N_{RPA,30-40min}$) between 150 mM NaCl and 15 mM NaCl conditions. Similar to the experimental intensity analysis, the results of comparison in simulation suggest that the RPA amounts loaded during 30-40 min are basically comparable at both salt concentrations, though with some small deviations.

b @ 40-min

Length (nt)	5,000		5,000		5,000		5,000		5,000	
Fold	0		1		4		10		25	
NaCl [mM]	150	15	150	15	150	15	150	15	150	15
$\Delta N_{RPA,30-40min}$	0	0	10	13	33	42	64	71	98	98
p-value	0.8553		<0.0001		<0.0001		<0.0001		0.2850	

Reviewer Only Figure 7 (included into Supplementary Fig. 6b). Simulated RPA loaded number during 30-40 min. (a) Representative kymographs of 100 pM RPA-WT-MeGFP with 150 mM NaCl and its length analysis. The ssDNA-RPA complex at start point was obtained by pre-incubation with excess RPA. (b) Schematic of RPA binding curve and mode transition.

6. Fig 6c: results of Δ -WH at 15 mM NaCl should be included if available.

We thank the reviewer for the suggestion. We have performed the ssDNA Curtains experiments suggested by the reviewer and have now included the data into Fig. 6b-c and Supplementary Fig. 7b-c. Consistent with the role of Δ -WH in facilitating the 20-nt mode, reducing the salt concentration to 15 mM NaCl did not alter the length and intensity changes of RPA- Δ WH with 10-fold and 25-fold RPA, though with a slightly slower kinetics of 25-fold RPA- Δ WH at 15 mM NaCl, which suggest the potential contribution of intramolecular interactions outside of Rfa2-WH in regulating the RPA ssDNA binding modes.

7. There are grammatical errors throughout the manuscript. Please proof the revised manuscript carefully before resubmission.

We thank the reviewer for the comments and suggestions. We have carefully proofed the manuscript thoroughly to avoid typo and errors, and made relevant revision to the text.

Reviewer #3 (Remarks to the Author):

RPA is a heterotrimeric ssDNA binding protein that plays important roles in all aspects of DNA metabolism involving an ssDNA intermediate, including the repair of DNA damage

through homologous recombination (HR). An interesting problem with RPA is that in cells, RPA is one of the first proteins to bind to early ssDNA intermediates during HR and it binds to ssDNA much more tightly than the recombinase Rad51, which is the key protein necessary for catalyzing the DNA strand exchange reactions that take place in HR. Full mechanistic details of how a relatively weak ssDNA binding protein (Rad51) manages to replace a tight ssDNA binding protein (RPA) are lacking within the field. In cells this process requires the involvement of mediator proteins such as Rad52 (in yeast) or BRCA2 (in humans), but how they accomplish this replacement reaction is not really understood other than at a very basic level.

In this study, Ding et al. use a combination of bulk biochemical, single molecule and computational simulations to investigate the properties of RPA-ssDNA complexes to try to establish how their biophysical properties might facilitate the early stages of Rad51 filament formation. The result of this work is a new quantitative model that describes how the dynamic transitions between distinct RPA ssDNA-binding modes can themselves contribute to Rad51 filament formation, and how the mediator protein Rad52 can assist in this process. The authors propose that RPA exists in a “protection mode” that enables it to protect ssDNA from potentially damaging nucleases and a “action mode” allowing for its replacement by Rad51.

The experiments are well designed, the data are technically good, the results in general support the authors model (with some caveats) and will also generate interest within the field. The manuscript does however require extensive English language editing before it can be published. My scientific comments (below) are relatively minor and focused more on some clarifications.

We thank the reviewer for these important comments and questions, and we are glad to hear that the reviewer appreciates the complexity of the system we chose to study. New simulations have been conducted, and the relevant text and references have been revised

according to the reviewer's suggestions (see below). We also added some of these thoughts inspired by the reviewer into the discussion part in the revised manuscript. All changes in the manuscript text file were highlighted in red.

Comments:

Figure 2: Most readers will not know the significance of panel iii in figure 2d. This should be explained in the main text or in the figure legend. In figure 2e panel ii, the PI will need to clarify that the trace represents just one ssDNA molecule, and it would be beneficial if the error bars were shown (in the legend, it states that "Error bars, mean \pm s.e.m., but I don't see any error bars). Be sure to explain that in subsequent figures the traces represent individual ssDNA molecules, and the boxplots represent the collective data.

We thank the reviewer for the important comments and suggestions. We are sorry for the confusion. We have revised the relevant text to emphasize the significance of turning off flow in Fig. 2d(iii) and to clarify the statistical meaning of the curve in Fig. 2e(ii). The curve in Fig. 2e(ii) represents the collective data of many ssDNA molecules. We are sorry that the error bars in Fig. 2e(ii) were previously shown in an obscure gray color. We have now revised the Fig. 2e(ii) to use a more visible color for the error bars.

Figure 3: The side-by-side presentation of the data sets in Figure 3c and 3d seem a bit confusing. Is there some way to emphasize that both bars represent the same Fold RPA? This also applies to other figures with boxplots.

We thank the reviewer for the comments and suggestions. We are sorry for the confusion. To make it clearer, we have revised the Fig. 3c-d and all the boxplots in subsequent figures by adding borders between different Fold boxes.

Regarding the simulations, the y-axes are labeled as "normalized length increments", which I presume to be derived from the value of $x_r(t)$, and the resulting simulation data is intended to match the presentation of the single molecule experiments. It is not clear to

me how the simulation length increments account for each of the three possible states of the bound ssDNA (naked ssDNA, RPA in the 20-nt mode and RPA in the 30-nt mode). My impression is that each of these states must have an assigned length value within the simulation and that they must also account for the fact that they are being compared to ssDNA molecules held in an extended configuration within a lamina flow system. If my interpretation is correct, what are the mean extended length values for each aforementioned state? I suppose this corresponds to the “m” values in Figure 4a, correct? If so, what are the numerical values for “m”, and do they make physical sense with respect to ssDNA characteristics in the flow stretched system?

We thank the reviewer for the comments and questions. The reviewer has made a correct interpretation for the length analysis principle in our simulations.

Basically, we assigned a length value to each state of ssDNA just as the reviewer said. As shown in Fig. 4a, the unit length of naked ssDNA is “m” nm per nucleotide. Since we assume RPA in different ssDNA binding modes have different length extension effect to ssDNA, we set two length scaling factors α and β for 30-nt mode and 20-nt mode respectively. Thus, the length of the 30-nt ssDNA covered by RPA in 30-nt mode should be $\alpha \cdot 30 \cdot m$ nm, and for 20-nt mode, $\beta \cdot 20 \cdot m$ nm. With the length values for the three possible states of ssDNA bound by RPA assigned, we can now calculate the total extension length of the whole ssDNA bound by many RPA molecules in different states. While in DNA Curtains analyses, we cannot directly compare the abstract length increment between different ssDNA-RPA complex because of the heterogeneity in length of RCR products. But since we can track the length of each ssDNA molecules at any moments ($L_{(t)}$), we can then calculate the “normalized length” as $L_{(t)}/L_{(30\ min)}$ for each ssDNA-RPA complex to compare the ssDNA extension length. To fit the experimental results, we also calculated the “normalized length” from simulation results:

$$L(t)/L_{(30 \text{ min})} = \frac{(n_{20\text{-nt mode}(t)} \cdot \beta \cdot 20 \cdot m + n_{30\text{-nt mode}(t)} \cdot \beta \cdot 30 \cdot m + n_{\text{naked ssDNA}(t)} \cdot m)}{(n_{20\text{-nt mode}(30 \text{ min})} \cdot \beta \cdot 20 \cdot m + n_{30\text{-nt mode}(30 \text{ min})} \cdot \beta \cdot 30 \cdot m + n_{\text{naked ssDNA}(30 \text{ min})} \cdot m)} = \frac{(n_{20\text{-nt mode}(t)} \cdot \beta \cdot 20 + n_{30\text{-nt mode}(t)} \cdot \beta \cdot 30 + n_{\text{naked ssDNA}(t)})}{(n_{20\text{-nt mode}(30 \text{ min})} \cdot \beta \cdot 20 + n_{30\text{-nt mode}(30 \text{ min})} \cdot \beta \cdot 30 + n_{\text{naked ssDNA}(30 \text{ min})})}, \text{ where "m" can be eliminated.}$$

Therefore, our simulation model can determine the fold extension of ssDNA by RPA in 30-nt mode (α) and in 20-nt mode (β), but does not need to provide a numeric value for "m".

Nevertheless, it is very important to discuss the physical sense for the theoretical simulated ssDNA length. According to the estimation of Ilya J. Finkelstein and his colleagues (2018)⁴, the length of ssDNA held by 0.4 mL/min (0.4 pN) flow rate within a lamina flow system is estimated to be 0.18 nm/nt. They also added saturated hRPA (5-10 nM) and found the bound ssDNA was stretched by 1.84-fold. According to our simulation results, RPA in 30-nt and 20-nt mode can stretched the ssDNA by 1.5-fold and 2.4-fold, respectively, at 150 mM NaCl. With these numbers, we can now calculate the estimated extension length of ssDNA bound by RPA in different binding modes: for ssDNA covered by RPA in 20-nt mode, 8.6 nm, and for ssDNA covered by 30-nt mode, 8.1 nm. While there is no available structure of ssDNA bound by yeast RPA protomer, Nikola P. Pavletich and his colleagues (2012) had reported the crystal structure of ssDNA bound by *U. maydis* RPA in 30-nt mode¹⁸, in which the ssDNA was stretched to 5.5 nm in linearized direction. The length measured from the crystal structure is comparable to our measurement though with 2.6 nm shorter, possibly caused by the crystal packing effect. Overall, the simulation results make physical sense with respect to our current knowledge of ssDNA characteristics. Hopefully these inferences could answer the questions of the reviewer.

An important potential issue with the authors' model is that it assumes that there is an absolute requirement for an ≥ 18 -nt ssDNA gap to allow for the initial nucleation of a Rad51 filament and it does not consider the possibility that shorter ssDNA gaps may allow for nucleation events wherein further addition of nucleating monomers promotes

the release or partial release of RPA ssDNA binding domains. For example, a 15 nt gap would allow for 5 Rad51 monomers, and a 6th could easily bind if one of the RPA OB-folds transiently dissociated from the ssDNA, or if RPA were to diffuse a short distance along the ssDNA (transient domain dissociation and 1D diffusion are both known characteristics of RPA). Similar arguments could be made for even smaller gaps. These features are not accounted for in the computer simulations or in the authors' model. I do not think that these issues invalidate the authors' general model (i.e., transitions between distinct RPA binding modes may affect the fraction of ssDNA that is accessible for binding by other proteins), nor do I think they need to be experimentally or computationally addressed at this stage (probably too difficult), but they should at least be clear acknowledged in the Discussion.

We thank the reviewer for the important comments and questions on the possible dynamic behaviors in this system. We are sorry for the confusion. We have revised relevant text and added relevant discussions inspired by the reviewer to better elucidate the dynamic properties of the ssDNA gaps formed between neighboring RPA molecules in our model.

Although we set an absolute threshold for an ≥ 18 -nt ssDNA gap to allow for the assembly of a Rad51 nucleofilament in the simulation model, the nucleation events on smaller ssDNA gaps were also possible in this model. We note that the size of the ssDNA gaps transiently formed between two adjacent RPA molecules was highly dynamic, due to the dissociation or binding event of the RPA protomer or a single OB-fold. In our model, a 15-nt ssDNA gap can easily turn into ≥ 18 -nt in two possible ways: first, one RPA dissociates from the ssDNA and then another RPA binds to ssDNA ≥ 3 -nt aside; second, the OB-D domain of the 5'-RPA bound in 30-nt mode dissociates from the ssDNA.

To answer the question of diffusion, we have added diffusion to our model and conducted new simulations for the 1D diffusion of RPA along the DNA. We found that

adding diffusion to our model did not significantly alter the binding dynamics of RPA or the size distribution of ssDNA gaps.

In short, we model it as a symmetric 1D random walk with simple exclusion effects. We noted that the diffusion coefficient for hRPA has been measured to be $> 10^3$ nt²/s and we did not find any literatures reporting diffusion coefficient for scRPA¹³. However, fast diffusion makes the exact Gillespie simulation intractable.

Fortunately, adding diffusion to our model does not significantly affect the dynamics of the RPA-DNA interaction, as shown in Reviewer Only Fig. 8. We first conduct a simulation with varying diffusion coefficients from 10^{-6} nt²/s to 10^0 nt²/s, while keeping other parameters the same (Reviewer Only Fig. 8a). We have found that the L2 norm of the simulated traces is insensitive to the diffusion coefficient. Furthermore, we reduced the total length of DNA to 100 nt and experimented with larger diffusion coefficients up to 10^2 nt²/s (Reviewer Only Fig. 8b). The same level of insensitivity was observed again.

Reviewer Only Figure 8. The difference from experimental value (loss function, L2 norm) of the simulated traces is insensitive to the diffusion coefficient. (a) The diffusion coefficient is varied from 10^{-6} nt²/s to 10^0 nt²/s, while keeping other parameters the same as the ones obtained by the gradient descent method. **(b)** We further increase the diffusion coefficient to 10^2 nt²/s and reduce the total length of DNA to 500bp. The L2 norm of the simulated traces is still insensitive to the diffusion coefficient.

Given that the diffusion around 10^2 nt²/s is much faster than the binding and conformation switching rates in that system, we therefore conclude that even in the fast diffusion regime, the dynamics of the RPA-DNA interaction is similar to the slow diffusion regime.

This conclusion does not negate the role of diffusion and can possibly be interpreted as following. The primary role of diffusion is to change the position of a DNA-bound RPA. However, unbinding and rebinding of an RPA is equivalent to diffusion in terms of moving RPA around. Consequently, introducing diffusion does not introduce new physics into our model. Since the diffusion coefficient of yeast RPA remains unknown, we chose not to include this data in the revised manuscript and will revisit this in future studies.

References

- 1 Gibb, B. *et al.* Concentration-dependent exchange of replication protein A on single-stranded DNA revealed by single-molecule imaging. *Plos One* **9**, e87922, doi:10.1371/journal.pone.0087922 (2014).
- 2 Kim, C. S., Paulus, B. F. & Wold, M. S. Interactions of Human Replication Protein-a with Oligonucleotides. *Biochemistry-Us* **33**, 14197-14206, doi:DOI 10.1021/bi00251a031 (1994).
- 3 Lisby, M., Barlow, J. H., Burgess, R. C. & Rothstein, R. Choreography of the DNA damage response: Spatiotemporal relationships among checkpoint and repair proteins. *Cell* **118**, 699-713, doi:DOI 10.1016/j.cell.2004.08.015 (2004).
- 4 Schaub, J. M., Zhang, H., Soniat, M. M. & Finkelstein, I. J. Assessing Protein Dynamics on Low-Complexity Single-Stranded DNA Curtains. *Langmuir* **34**, 14882-14890, doi:10.1021/acs.langmuir.8b01812 (2018).
- 5 Gillespie, D. T. Exact Stochastic Simulation of Coupled Chemical-Reactions. *J Phys Chem-Us* **81**, 2340-2361, doi:DOI 10.1021/j100540a008 (1977).
- 6 Kumaran, S., Kozlov, A. G. & Lohman, T. M. Saccharomyces cerevisiae Replication Protein A Binds to Single-Stranded DNA in Multiple Salt-Dependent Modes†. *Biochemistry* **45**, 11958-11973, doi:10.1021/bi060994r (2006).
- 7 Kim, C. & Wold, M. S. Recombinant Human Replication Protein-a Binds to Polynucleotides with Low Cooperativity. *Biochemistry-Us* **34**, 2058-2064, doi:DOI 10.1021/bi00006a028 (1995).
- 8 Mitsis, P. G., Kowalczykowski, S. C. & Lehman, I. R. A single-stranded DNA binding protein from Drosophila melanogaster: characterization of the heterotrimeric protein and its interaction with single-stranded DNA. *Biochemistry* **32** **19**, 5257-5266 (1993).
- 9 Sugiyama, T., Zaitseva, E. M. & Kowalczykowski, S. C. A Single-stranded DNA-binding Protein Is Needed for Efficient Presynaptic Complex Formation by the Saccharomyces cerevisiae Rad51 Protein*. *Journal of Biological Chemistry* **272**, 7940-7945, doi:10.1074/jbc.272.12.7940 (1997).
- 10 Bastin-Shanower, S. A. & Brill, S. J. Functional Analysis of the Four DNA Binding Domains of Replication Protein A. *Journal of Biological Chemistry* **276**, 36446-36453, doi:10.1074/jbc.m104386200 (2001).
- 11 Sibenaller, Z. A., Sorensen, B. R. & Wold, M. S. The 32- and 14-Kilodalton Subunits of Replication Protein A Are Responsible for Species-Specific Interactions with Single-Stranded DNA. *Biochemistry* **37**, 12496-12506, doi:10.1021/bi981110+ (1998).
- 12 Yates, L. A. *et al.* A structural and dynamic model for the assembly of Replication Protein A on single-stranded DNA. *Nature Communications* **9**, doi:10.1038/s41467-018-07883-7 (2018).
- 13 Nguyen, B. *et al.* Diffusion of Human Replication Protein A along Single-Stranded DNA. *J. Mol. Biol.* **426**, 3246-3261, doi:10.1016/j.jmb.2014.07.014 (2014).
- 14 Pokhrel, N. *et al.* Dynamics and selective remodeling of the DNA-binding domains of RPA. *Nat Struct Mol Biol* **26**, 129-136, doi:10.1038/s41594-018-0181-y (2019).
- 15 Ma, C. J., Gibb, B., Kwon, Y., Sung, P. & Greene, E. C. Protein dynamics of human

- RPA and RAD51 on ssDNA during assembly and disassembly of the RAD51 filament. *Nucleic Acids Research* **45**, 749-761, doi:10.1093/nar/gkw1125 (2017).
- 16 Gibb, B., Silverstein, T. D., Finkelstein, I. J. & Greene, E. C. Single-Stranded DNA Curtains for Real-Time Single-Molecule Visualization of Protein–Nucleic Acid Interactions. *Analytical Chemistry* **84**, 7607-7612, doi:10.1021/ac302117z (2012).
- 17 Bastin-Shanower, S. A. & Brill, S. J. Functional analysis of the four DNA binding domains of replication protein A - The role of RPA2 in ssDNA binding. *J Biol Chem* **276**, 36446-36453, doi:DOI 10.1074/jbc.M104386200 (2001).
- 18 Fan, J. & Pavletich, N. P. Structure and conformational change of a replication protein A heterotrimer bound to ssDNA. *Gene Dev* **26**, 2337-2347, doi:10.1101/gad.194787.112 (2012).

REVIEWERS' COMMENTS

Reviewer #2 (Remarks to the Author):

The authors have addressed every comment thoughtfully.

The study is impactful for understanding how RPA dynamics regulate the assembly of Rad51 complexes on ssDNA. I am supportive of its acceptance.

Reviewer #3 (Remarks to the Author):

The authors have addressed all of my prior comments and I believe that the manuscript is now ready for publication.

Final revisions for Nature Communications manuscript NCOMMS-22-48278A

May 27, 2023

REVIEWERS' COMMENTS

Reviewer #1:

N/A.

Reviewer #2 (Remarks to the Author):

The authors have addressed every comment thoughtfully.

The study is impactful for understanding how RPA dynamics regulate the assembly of Rad51 complexes on ssDNA. I am supportive of its acceptance.

We thank the reviewer's positive comments.

Reviewer #3 (Remarks to the Author):

The authors have addressed all of my prior comments and I believe that the manuscript is now ready for publication.

We thank the reviewer's positive comments.